Eocene Loranthaceae pollen pushes back divergence ages for major splits in the family

http://orcid.org/0000-0002-1874-6412 Grímsson Friðgeir 1 fridgeir.grimsson@univie.ac.at
Kapli Paschalia 2
Hofmann Christa-Charlotte 1
http://orcid.org/0000-0002-0220-6921 Zetter Reinhard 1
http://orcid.org/0000-0003-0674-3553 Grimm Guido W. 1 3 grimmiges@gmail.com
1 Department of Palaeontology, University of Vienna , Wien , Austria
2 The Exelixis Lab, Scientific Computing Group, Heidelberg Institute for Theoretical Studies , Heidelberg , Germany
3 Orléans , France
De Baets Kenneth
Electronic publication date: 2017 Jun 7
Publication date: 2017
Volume: 5
Electronic Location ID: e3373
Received 2016 Dec 16; Accepted 2017 May 4
Copyright: © 2017 Grímsson et al.
Copyright year: 2017
Copyright holder: Grímsson et al.
License: This is an open access article distributed under the terms of the Creative Commons Attribution License, which permits unrestricted use, distribution, reproduction and adaptation in any medium and for any purpose provided that it is properly attributed. For attribution, the original author(s), title, publication source (PeerJ) and either DOI or URL of the article must be cited.
License URL: https://creativecommons.org/licenses/by/4.0/

Keywords: Pollen morphology, Pollen as minimum age priors, Uncorrelated clock node dating, Topological uncertainty, Palaeophytogeography, Lineage-through-time plot, Santalales

Funding: Austrian Science Fund (FWF) P24427-B25 and M1751-B16 Synthesis FP7–the European Union-funded Integrated Activities DK-TAF 1971, SE-TAF 1918, and GB-TAF 3740 This study was funded by the Austrian Science Fund (FWF) with grants to FG, project number P24427-B25, and GWG, project number M1751-B16, and by Synthesis FP7–the European Union-funded Integrated Activities Grants DK-TAF 1971, SE-TAF 1918, and GB-TAF 3740 to FG. The funders had no role in study design, data collection and analysis, decision to publish, or preparation of the manuscript.

==============================
Background

We revisit the palaeopalynological record of Loranthaceae, using pollen ornamentation to discriminate lineages and to test molecular dating estimates for the diversification of major lineages.

Methods

Fossil Loranthaceae pollen from the Eocene and Oligocene are analysed and documented using scanning-electron microscopy. These fossils were associated with molecular-defined clades and used as minimum age constraints for Bayesian node dating using different topological scenarios.

Results

The fossil Loranthaceae pollen document the presence of at least one extant root-parasitic lineage (Nuytsieae) and two currently aerial parasitic lineages (Psittacanthinae and Loranthinae) by the end of the Eocene in the Northern Hemisphere. Phases of increased lineage diversification (late Eocene, middle Miocene) coincide with global warm phases.

Discussion

With the generation of molecular data becoming easier and less expensive every day, neontological research should re-focus on conserved morphologies that can be traced through the fossil record. The pollen, representing the male gametophytic generation of plants and often a taxonomic indicator, can be such a tracer. Analogously, palaeontological research should put more effort into diagnosing Cenozoic fossils with the aim of including them into modern systematic frameworks.

Introduction

The Loranthaceae (order Santalales), a moderately large family comprising about 76 genera and over 1,000 species in five tribes (Nickrent, 1997 onwards; Nickrent et al., 2010), has a wide geographical distribution. Today, there is a relatively clear geographic split between a New World group (Psittacanthinae) and Old World-Australasian lineages (Elythrantheae and Lorantheae), which gave rise to the hypothesis that the initial Loranthaceae diversification was linked to the final phase of the Gondwana breakup in the Late Cretaceous (e.g. Barlow, 1990; Vidal-Russell & Nickrent, 2007). Only three of the more than 70 genera are root parasites and the rest are aerial branch parasites. Molecular studies on Loranthaceae (and Santalales in general) have thus focused on three issues: (1) clarifying the evolutionary relationships within the family (Vidal-Russell & Nickrent, 2008a); (2) explaining the transition from root to aerial parasitism (Wilson & Calvin, 2006); (3) dating the time of transition to aerial parasitism (Vidal-Russell & Nickrent, 2008b). All molecular studies using outgroups recognised the south-western Australian, root-parasitic, monotypic Nuytsia R. Br. (monogeneric tribe Nuytsieae; Nickrent et al., 2010) as the first diverging lineage in the family (Wilson & Calvin, 2006; Vidal-Russell & Nickrent, 2008a; Su et al., 2015). The other two Loranthaceae root parasites (Atkinsonia F. Muell., Gaiadendron G. Don; tribe Gaiadendreae) formed a grade to the New World aerial parasites (Wilson & Calvin, 2006; multiple origins of aerial parasitism) or all aerial parasitic genera of the family (Vidal-Russell & Nickrent, 2008a; Vidal-Russell & Nickrent, 2008b; Su et al., 2015; singular origin). Using a time-calibrated phylogeny, Vidal-Russell & Nickrent (2008b) concluded that Loranthaceae diverged from other Santalales lineages in the uppermost Cretaceous. The first radiation—the divergence of root parasites Nuytsia, Atkinsonia and Gaiadendron—was during the Eocene. The crown age of the aerial parasitic clade within the Loranthaceae, comprising the mostly New World Psittacantheae and Old World-Australasian Erytrantheae and Lorantheae, was placed in the middle Oligocene, approximately 28 Ma (estimated via a Bayesian relaxed clock and fixing the Santalales root to a maximum age of 114 Ma); a time characterised by global cooling (Zachos et al., 2001) and retreat of subtropical and tropical vegetation.

Although molecular and morphological interrelationships of Loranthaceae genera are considered now to be relatively clear (Nickrent et al., 2010; Su et al., 2015; but see Grímsson, Grimm & Zetter, 2017), the timing of divergence between the genera has not been cross-checked with available evidence from the fossil record (e.g. Muller, 1981; Song, Wang & Huang, 2004; Macphail et al., 2012). Also, the phytogeographic history of the family is based merely on the present distribution of its genera (e.g. Vidal-Russell & Nickrent, 2007) and has not yet been explored in detail (Vidal-Russell & Nickrent, 2008a, p. 1027). The latest hypothesis put forward was that Loranthaceae originated when South America, Antarctica and Australia were still connected, and that two large-scale migration events, one from New Zealand and one from Australia, shaped the modern distribution (Vidal-Russell & Nickrent, 2008a, 2008b). The single shift to aerial parasitism was estimated to be of middle Oligocene age. Thus, older fossil records, the oldest going back to the early Eocene (c. 50 Ma) in Australia, were considered to represent root parasites or extinct clades of aerial parasites (Macphail et al., 2012).

The outstanding work on the pollen morphology of extant Loranthaceae by Feuer & Kuijt (1979, 1980, 1985) and other Santalales lineages (Maguire, Wurdack & Huang, 1974; Feuer, 1977, 1978, 1981; Feuer & Kuijt, 1978, 1982; Feuer, Kuijt & Wiens, 1982) demonstrated that most pollen produced by members of the Loranthaceae cannot be confused with pollen from other angiosperm families (Grímsson, Grimm & Zetter, 2017). Grímsson, Grimm & Zetter (2017) distinguished four general types (Pollen Type A, B, C, D), of which only one (Pollen Type A) could be confused with pollen of other Santalales lineages, and would unlikely be recognised as Loranthaceae pollen if found in a fossil pollen sample. Combined application of light microscopy (LM), scanning-electron microscopy (SEM), and transmission electron microscopy (TEM) revealed that pollen morphologies—including the many variants of B-type pollen—are conserved at various taxonomic levels within Loranthaceae (Feuer & Kuijt, 1978, 1979, 1980, 1985; Caires, 2012; Grímsson, Grimm & Zetter, 2017). Thus, dispersed fossil pollen can aid in the reconstruction of past distributions of Loranthaceae lineages and shed light on the timing of the origin of the modern clades. Being the male gametophyte of a plant, pollen are generally conserved in morphology. Diagnostic (lineage-specific) pollen hence allow for the tracing of modern lineages deep into the past (e.g. Zetter, Hesse & Huber, 2002; Grímsson, Zetter & Hofmann, 2011; Grímsson et al., 2016).

Here, we describe new fossil Loranthaceae pollen grains from the middle Eocene of the United States, Greenland, Central Europe, and East Asia, and from the late Oligocene/early Miocene of Germany. The diagnostic morphological features of the pollen provided sufficient details to assign the fossil pollen to distinct lineages within the Loranthaceae. These fossil pollen represent the earliest unambiguous reports of the root parasitic Nuytsieae, and the presently aerial parasitic lineages Psittacanthinae, Elytrantheae and Lorantheae. Thus, they can be used as potential ingroup minimum age priors for node dating, and to refine our knowledge about the evolutionary history of the Loranthaceae.

Materials and Methods

Origin of samples and geological background

The fossil Loranthaceae pollen identified during this study occurred in six different sedimentary rock samples: (1) the Claiborne Group of the Miller Clay Pit in Henry County, Tennessee, United States (sample UF 15817-062117); (2) the Hareøen Formation (middle Eocene) on Qeqertarsuatsiaq Island (Hareøen), western Greenland; (3) the Borkener coal measures of the Stolzenbach underground coal mine, near Kassel, Germany; (4) the Profen Formation (middle Eocene) of the Profen opencast mine, close to Leipzig in Germany; (5) the Changchang Formation (middle Eocene) on northern Hainan Island, South China; (6) the Melker Series of the NÖ05 borehole positioned close to Theiss, near Krems, Lower Austria; and (7) the Cottbus/Spremberg Formations (late Oligocene/early Miocene) of Altmittweida in Saxony, Germany (Table 1). For details on the geographic positions, geology, palaeoecology, and previously known fossil plants from these formations and localities see Table 1 and references therein. Epoch names and ages in Table 1 follow Cohen et al. (2013 [updated]).

Table 1 Information on sample sites.

	Miller Clay Pit MT1–MT3	Aamaruutissaa MT	Stolzenbach MT	Profen MT1–MT5	Changchang MT	Theiss MT	Altmittweida MT	
Location	Miller Clay Pit, Henry County, Tennessee, United States	Aamaruutissaa, southeast Hareøen Island, western Greenland	Stolzenbach underground coalmine, Kassel, Germany	Profen opencast mine, close to Leipzig, Germany	Changchang Basin, close to Jiazi Town, Qiongshan County, Hainan, China	Theiss, borehole southeast of Krems, Lower Austria	Altmittweida, Saxony, Germany	
Latitude and longitude (ca.)	36°13′N, 88°27′W	70°24′N, 54°41′W	51°0′N, 9°17′E	51°09′N, 12°11′E	19°38′N, 110°27′E	48°23′N, 15°41′E	50°58′N, 12°55′E	
Lithostratigraphy	Claiborne Group	Hareøen Formation	Borkener coal measures	Profen Formation	Changchang Formation	Melker Series	Cottbus/Spremberg Formations	
Epoch*	Lutetian	Late Lutetian-early Bartonian	Lutetian	Bartonian	Lutetian-Bartonian	Rupelian	Chattian to Aquitanian	
Age (Ma)*	47.8–41.2*	42–40 [absolute dating]	47.8–41.2*	41.2–38*	47.8–37.8*	33.9–28.1*	28.1–20.44*	
According to	Litho- and biostratigraphy	Chrono-, litho- and biostratigraphy	Litho- and biostratigraphy	Litho- and biostratigraphy	Litho- and biostratigraphy	Litho- and biostratigraphy	Litho- and biostratigraphy	
Notes on palynofloras	Dominated by angiosperms; rich in Fagaceae, Juglandaceae, Sapotaceae, Anarcardiaceae, Olacaceae, Cannabaceae, and Altingiaceae	Diverse spore and pollen flora; rich in Cupressaceae and angiosperms; Fagus, Quercus and Castaneoideae pollen abundant	Dominated by angiosperms; rich in Ericaceae, Fagaceae, Hamamelidaceae, Altingiaceae, Combretaceae, Burseraceae, Icacinaceae, Juglandaceae, Lecythidiaceae, and Sapotaceae	Dominated by angiosperms; rich in Anacardiaceae, Araceae, Arecaceae, Fagaceae, Sapotaceae, Symplocaceae, and Combretaceae	Diverse in angiosperms; rich in Fagaceae pollen, especially Quercus and Castaneoideae	Dominated by angiosperms, rich in Fagaceae, Sapotaceae, Juglandaceae, Vitaceae, Malvaceae, Symplocaceae, Cornaceae, Oleaceae, and Arecaceae	Diverse in angiosperms; rich in Juglandaceae and Fagaceae genera	
For further info on the geological background, stratigraphy [S], palaeoenvironment, palaeoclimate, and plant fossils [P]	Tschudy (1973[P]); Potter (1976[P]); Taylor (1989[P]); Dilcher & Lott (2005[P/S]); Wang, Blanchard & Dilcher (2013[P])	Heer (1883[P]); Hald (1976, 1977[S]); Schmidt et al. (2005[S]); Dam et al. (2009[S]); Grímsson et al. (2014a, 2015); Larsen et al. (2015[S]); Manchester, Grímsson & Zetter (2015[P])	Oschkinis & Gregor (1992[P/S]); Gregor (2005[P]); Hottenrott, Gregor & Oschkinis (2010[P]); Gregor & Oschkinis (2013[P]); Manchester, Grímsson & Zetter (2015[P])	Krutzsch & Lenk (1973[P/S]); Pälchen & Walter (2011[S]); Manchester, Grímsson & Zetter (2015[P])	Guo (1979[P]); Lei et al. (1992[P]); Jin et al. (2002[P]); Yao et al. (2009[P]); Spicer et al. (2014[S/P])	Hochuli (1978[P]); Weber & Weiss (1983[S]); Eschig (1992[P/S]); Grímsson, Ferguson & Zetter (2012[P])	Engelhardt (1870[P]); Mai & Walther (1991[P]); Standke (2008[S]); Kmenta (2011[P]); Kmenta & Zetter (2013[P])	
Note:

* Following Cohen et al. (2013, update).

Preparation of samples

The sedimentary rock samples were processed according to the protocols outlined in Grímsson, Denk & Zetter (2008). We investigated the fossil Loranthaceae pollen grains using the ‘single grain method’ (Zetter, 1989), whereby the same fossil pollen grain is first analysed under the LM and then SEM. SEM stubs produced under this study are stored in the collection of the Department of Palaeontology, University of Vienna, Austria, under accession numbers IPUW 7513/076-100.

Molecular framework

For molecular data we relied on a 2014 NCBI GenBank harvest compiled for an earlier study (Grímsson, Grimm & Zetter, 2017). Gene banks now (as of December 1st, 2016) include ∼100 additional accessions (File S2); but the majority of these are either microsatellite marker sequences or sequences of gene regions too variable, or with insufficient taxonomic coverage within the Loranthaceae, to be of any use; thus, we opted against updating the dataset harvested for the preceding study. All analysis files (sorted by steps) are included in the online supporting archive (OSA) in the Supplementary Information (File S1: Steps 1–3 of analysis pipeline).

Given the problems with signals in Loranthaceae sequence data (Grímsson, Grimm & Zetter, 2017: Files S1, S6), we used the following protocol to prepare data sets for phylogenetic inferences and molecular dating (a detailed description is provided in File S1). First, we performed single-gene maximum likelihood (ML) inferences for five candidate gene regions using the complete harvested data with RAxML v. 8.2.4 (Stamatakis, 2014). This was mainly done to cross-check for problematic accessions and to test the phylogenetic coherence of multiple accessions of the same species/genus. As a consequence, we eliminated several more sequences, in addition to the ones not considered earlier, for computing strict genus-consensus sequences (see File S1, an emended version of Grímsson, Grimm & Zetter, 2017: File S2). The second step was to consense and concatenate the unproblematic data: strict species-consensus sequences, i.e. sequences summarising the information of accessions attributed to a species, were computed with g2cef (Göker & Grimm, 2008) and concatenated with Mesquite v. 2.75 (Maddison & Maddison, 2011). The third and final step was the inference of single- and oligo-gene ML trees using RAxML; branch support was established using non-parametric bootstrapping with the number of necessary bootstrap replicates determined by the extended majority rule consensus bootstop criterion (Pattengale et al., 2009). Potentially conflicting signals were explored using bootstrap (BS) consensus networks (bipartition networks; Grimm et al., 2006), a special form of consensus networks (Holland & Moulton, 2003) generated with SplitsTree v. 14.2 (option ‘count’, Huson & Bryant, 2006) in which edge lengths are proportional to the frequency of the according split in the BS (pseudo)replicate sample. The complex signal and overall divergence in the molecular data calls for a probabilistic inference method (i.e. ML or Bayesian inference), and a means for establishing branch support that can reflect the robustness of (partly) conflicting signals from the data. Regarding the latter, non-parametric bootstrapping is more informative (conservative) than Bayesian-inferred posterior probabilities (PP). If a certain proportion of alignment patterns (e.g. 30%) support a split B that is in conflict with the dominant split A (supported by 70% of the alignment patterns), the BS support under ML (BSML) will be accordingly split in the optimal case (BSML ∼70 for A vs. BSML ∼30 for B). The PP may converge to 1 for A and 0 for B as the MCMC chain(s) optimise(s) towards the topology that best explains the complete data. For the example of Loranthaceae, it can be demonstrated that branches with PP ∼ 1 >> BSML in the tree of Su et al. (2015) relate to relationships supported only by one gene region (matK), which outcompetes conflicting, partly unambiguous signals from all other gene regions; the latter captured in the BS pseudoreplicate samples (Grímsson, Grimm & Zetter, 2017: File S6). Readers interested in the behaviour of PP in comparison to the BSML support values used here can find the according information in the Supporting Information (File S1: set-up, File S5: results; Bayesian sampled topology files are included in the OSA).

Clock-rooting

A recent re-analysis of available molecular data using genus-consensus sequences (Grímsson, Grimm & Zetter, 2017) failed to unambiguously resolve basal relationships in Loranthaceae as was the case in earlier studies using placeholder accessions (Wilson & Calvin, 2006; Vidal-Russell & Nickrent, 2008a; Su et al., 2015; see Grímsson, Grimm & Zetter, 2017: File S6, for a critical assessment of the Loranthaceae data included by Su et al.). The problem of topological ambiguity worsens for the species tree inferred here, in part due to data gaps (see Inferences and supplement to Grímsson, Grimm & Zetter, 2017). Due to issues regarding ambiguity of the deepest splits within the Loranthaceae and likely outgroup–ingroup long-branch attraction (Grímsson, Grimm & Zetter, 2017: File S6), we inferred an alternative, clock-based root (Huelsenbeck, Bollback & Levine, 2002) for the Loranthaceae tree using beast v. 1.8.2 (Drummond & Rambaut, 2007; Drummond et al., 2012), following the example of an earlier study on Acer (Renner et al., 2008). Clock-rooting was performed for five main datasets differing in the gene region coverage (all gene regions, all but excluding the most variable trnL intron, 5’ exon (can be incomplete) and trnL–trnF spacer (trnL/LF) region, only plastid regions including or excluding the trnL/LF, only nuclear regions). In addition, the taxon-reduced data set used for the final dating step was analysed (for further details see File S1). For each of the matrices we performed a beast run under partition specific substitution models, unconstrained tree topology, a Yule tree prior, and uncorrelated log-normal clock prior [ucld.mean ∼ Gamma (0.001, 1,000)]. The best fitting substitution models per partition, among the available in beast, were selected with JModelTest (Darriba et al., 2012). Each analysis was conducted for 2 × 107 generations with a sampling frequency of 10−3 (for further details see File S1) (Table 2; Step 4 of analysis pipeline).

Table 2 Results of the clock-rooting analyses.

Species set	Gene set	Inferred root	
All	All	Between Lorantheae core clade and all other Loranthaceae (including Loranthinae and Ileostylinae; not used as rooting scenario for subsequent analyses)	
All	All, excluding trnL/LF	Between Lorantheae and all other Loranthaceae (= rooting scenario 2)	
All	Nuclear ribosomal DNAs only	Between Lorantheae and all other Loranthaceae (= rooting scenario 2)	
All	Chloroplast regions only	Between Lorantheae and all other Loranthaceae (= rooting scenario 2)	
All	Chloroplast genes only	Between Lorantheae and all other Loranthaceae (= rooting scenario 2)	
Reduced	All	Between Nuytsia and all other Loranthaceae (equals outgroup inferred root; = rooting scenario 1)	

Basic setup of molecular dating

Nine of the 13 new described fossil pollen from the Eocene to Oligocene (see Results) were used as minimum age constraints (informing three to five node height priors per analysis) for traditional node dating using a Bayesian uncorrelated clock (UC) approach; analyses were performed with beast v. 1.8.2. Table 2 lists the age priors used for the analyses. Dating was done in two phases (for set-up details see File S1). With respect to the non-trivial matrix signals and the branch-lengths seen in the ML tree, rate shifts should be considered during dating. Hence, we chose to use the UC approach over other node-dating alternatives (e.g. Renner et al., 2008; Smith, Beaulieu & Donoghue, 2009; see e.g. Dornburg et al., 2012 for bias in the case of mammal mtDNA) (Table 2; Steps 5 and 6 of analysis pipeline).

In the initial phase (Step 5), we inferred dated species phylogenies based on the complete concatenated data set for three rooting scenarios: (1) the commonly accepted root placing Nuytsia as sister to all other Loranthaceae (Vidal-Russell & Nickrent, 2008a; Nickrent et al., 2010; Su et al., 2015); (2) a clock-inferred root recognising the predominately Old World Lorantheae as sister to a mainly southern hemispheric clade that includes all three root parasites, the Psittacantheae and Elytrantheae (see Results); and (3) recognising Tupeia as sister to all other Loranthaceae. The 3rd scenario is based on the hypothesis that the typical oblate, ± triangular Loranthaceae pollen (Pollen Type B in Grímsson, Grimm & Zetter, 2017, a pollen type unique within the Santalales) evolved only once. The monotypic Tupeia is one of two Loranthaceae species with a spheroidal, echinate pollen as found in other Santalales lineages (Grímsson, Grimm & Zetter, 2017) and the only one sequenced so far. Irrespective of the data used, Tupeia is the taxon with the smallest root-tip distance within Loranthaceae, including trees rooted with Nuytsia (Su et al., 2015: Fig. 1B; Grímsson, Grimm & Zetter, 2017; this study).

Figure 1 LM micrographs (polar views) of all fossil Loranthaceae morphotypes.

(A) Miller Clay Pit MT1. (B) Miller Clay Pit MT1. (C) Miller Clay Pit MT2. (D) Miller Clay Pit MT3. (E) Aamarutissaa MT. (F) Stolzenbach MT. (G, H) Profen MT1. (I, J) Profen MT2. (K) Profen MT3. (L–O) Profen MT4. (P, Q) Profen MT5. (R, S) Changchang MT. (T–X) Theiss MT. (Y–Ä) Altmittweida MT.

For the final dating (Step 6), we used a taxon-reduced data set limited to 42 species covering all included gene regions to counter problems with missing data in the full data set. At this step we also included an alternative topology, which constrained the primary branching patterns seen in the tree of Su et al. (2015). This alternative topology was suggested by an anonymous reviewer reporting on the draft to Grímsson, Grimm & Zetter (2017) to depict the correct relationships between the major lineages and potentially early diverging, isolated, monotypic genera. Each analysis ran for 5 × 107 MCMC steps, under a similar set-up as described in the preceding step (clock-rooting). The tree was partially constrained each time to accommodate the placement of the fossils and the corresponding rooting hypothesis (see xml-setup files provided in the supplemental information [OSM], for details). Each analysis was run twice to ensure the runs converged to the stationary distribution. Finally, all age calibration priors (Table S1-1 in File S1) were modelled as normal distributions around the midpoint of the known time intervals (for further details see File S1). With respect to the variations in plant evolutionary rates (e.g. Guzmán & Vargas, 2010; Lockwood et al., 2013), we opted against performing any rate-based dating. Smith & Donoghue (2008) cautioned against the use of rate-based estimates of divergence times even when fossil calibration priors are lacking as it may lead to strong biases.

Descriptions

Some lineages (tribes, subtribes) and genera of modern Loranthaceae are characterised by unique pollen morphologies (autapomorphies in a strict Hennigian sense) or specific pollen character suites (Grímsson, Grimm & Zetter, 2017). Nevertheless, we refrained from using genus names to address the fossil pollen types described here—even if the pollen was highly similar or indistinguishable from a modern counterpart—for several reasons: (1) intra- and interspecific variation is not comprehensively understood in Loranthaceae; (2) the generic concepts in Loranthaceae are under on-going revision, (3) monotypic modern lineages/genera could have been more widespread and diverse in the past; and (4) occurrence of fossils combining features of two or more genera or lineages. Thus, all pollen grains are classified as morphotypes (MT) named after the locality where they were found (Figs. 1–6).

Figure 2 SEM micrographs of fossil Loranthaceae pollen similar to/intermediate between root parasites and Lorantheae and comparable extant pollen.

(A–D) Polar views of fossil pollen. (E–G) Polar views of extant pollen. (H–J) Close-ups of sculpturing in area of mesocolpium and along margo in fossil pollen. (K–M) Close-ups of sculpturing in area of mesocolpium and along margo in extant pollen. (A, H) Miller Clay Pit MT1. (B, I) Stolzenbach MT. (C, J) Profen MT1. (D) Theiss MT. (E, K) Nuytsia floribunda. (F, L) Gaiadendron punctatum. (G, M) Muellerina eucalyptoides. Scale bars: (A–M) = 1 μm.

Figure 3 SEM micrographs of fossil Loranthaceae pollen with affinity to Tripodanthus and extant pollen of the genus.

(A–D) Polar views of fossil pollen. (E, F) Polar views of extant pollen. (G–J) Close-ups of sculpturing in area of mesocolpium and along margo in fossil pollen. (K, L) Close-ups of sculpturing in area of mesocolpium and along margo in extant pollen. (A, G) Miller Clay Pit MT2. (B, C, H, I) Miller Clay Pit MT3. (D, J) Aamaruutissaa MT. (E, F, K, L) Tripodanthus acutifolius. Scale bars: (A–F) = 10 μm, (G–L) = 1 μm.

Figure 4 SEM micrographs of fossil Loranthaceae pollen with affinity to Elytrantheae and extant representatives.

(A–F) Polar views of fossil pollen. (G–I) Polar views of extant pollen. (A, B) Profen MT2. (C) Profen MT3. (D, E) Profen MT4. (F). Profen MT5. (G) Peraxilla tetrapetala. (H) Amylotheca sp. (I) Ligaria cuneifolia. Scale bar: (A–I) = 10 μm.

Figure 5 SEM micrographs of fossil Loranthaceae pollen with affinity to Elytrantheae and extant representatives.

(A–E) Close-ups of sculpturing in area of mesocolpium and along margo in fossil pollen. (F–H) Close-ups of sculpturing in area of mesocolpium and along margo in extant pollen. (A, B) Profen MT2. (C) Profen MT3. (D) Profen MT4. (E) Profen MT5. (F) Peraxilla tetrapetala. (G) Amylotheca sp. (H) Ligaria cuneifolia. Scale bar: (A–H) = 1 μm.

Figure 6 SEM micrographs of fossil Loranthaceae pollen with affinity to crown group Lorantheae and comparable extant pollen.

(A–D) Polar views of fossil pollen. (E–G) Polar views of extant pollen. (H–J) Close-ups of sculpturing in area of mesocolpium and along margo in fossil pollen. (K–M) Close-ups of sculpturing in area of mesocolpium and along margo in extant pollen. (A, B, H, I) Changchang MT. (C, D, J) Altmittweida MT. (E, K) Amyema gibberula. (F, L) Helixanthera kirkii. (G, M) Taxillus caloreas. Scale bars: (A–G) = 10 μm, (H–M) = 1 μm.

All fossil pollen described here falls within the variation of Pollen Type B according Grímsson, Grimm & Zetter (2017). Pollen grains of Type B are oblate (to various degrees), triangular to trilobate in polar view and show a ± psilate sculpturing in LM. They are basically syn(3)colpate, but also demisyn(3)colpate and zono(3)colpate (terminology follows Punt et al., 2007; see Grímsson, Grimm & Zetter, 2017: Fig. 1, for schematic drawings) in some genera/lineages. Usually, no further sculpturing details can be observed in LM except for occasional exine thickening or thinning at the pole (e.g. Figs. 1C, 1H, 1M, 1Y) and along the colpi or in the mesocolpium (e.g. Figs. 1C, 1D, 1R).

Miller Clay Pit MT1, aff. Nuytsia

Description

Pollen, oblate, concave triangular in polar view, no undistorted equatorial view available, equatorial apices obcordate, interapertural areas (mesocolpia) sunken; pollen small, equatorial diameter 15.0–18.3 μm in LM, 13.0–14.4 μm in SEM; zono(3)colpate, colpi long and narrow; exine 0.7–0.8 μm thick, nexine thinner than sexine; tectate; sculpturing psilate in LM, microechinate in area of mesocolpium in SEM, echini 0.3–0.8 μm long, 0.2–0.5 μm wide at base SEM; margo well developed, broad, psilate to partly granulate SEM; colpus membrane not observed (Figs. 1A, 1B, 2A, 2H; Plate S01, S02 in File S3).

Locality

Miller Clay Pit, Henry County, Tennessee, USA (Table 1).

Remarks

This pollen type is very similar to pollen of the extant southwestern Australian Nuytsia floribunda (Labill.) G. Don as figured by Feuer & Kuijt (1980) and Grímsson, Grimm & Zetter (2017); a pollen readily distinct from all other modern Loranthaceae. The fossil pollen only differs from Nuytsia by being zonocolpate and showing sunken (infolded) mesocolpia in LM and SEM. The shift from the basic syncolpate organisation to zonocolpate can be observed in several lineages of (modern) Loranthaceae. With respect to the high genetic distinctness of Nuytsia from all other Loranthaceae, the modern species likely represents the sole survivor of an early diverged lineage of root parasitic loranths. Hence, it is likely that ancestral or extinct members of Nuytsia/Nuytsieae had more morphological variation than can be observed in the sole surviving species.

Use as age constraint

The Miller Clay Pit MT1 can be used to constrain the root age of the lineage leading to Nuytsia, i.e. the Nuytsieae root age. Based on the currently available molecular data, the relationship of Nuytsia to the remainder of the genus and the other two extant root parasites is unclear. Nevertheless, Nuytsia is likely the sole modern-day representative of an early diverging lineage. For rooting scenario 1 (outgroup-inferred root) Miller Clay Pit MT1 serves as minimum age constraint for the MRCA of all (extant) Loranthaceae.

Miller Clay Pit MT2, aff. Tripodanthus

Description

Pollen, oblate, concave-triangular in polar view, no undistorted equatorial view available, equatorial apices T-shaped; pollen small, equatorial diameter 18.3–21.7 μm in LM, 17.9–20.2 μm in SEM; syn(3)colpate, colpi narrow; exine 1.2–1.5 μm thick, nexine thinner than sexine, intercolpial nexine thickening at pole, sexine thickened in area of mesocolpium (LM); tectate; sculpturing psilate in LM, (micro)baculate in area of mesocolpium in SEM, (micro)bacula densely packet, 0.2–0.9 μm long, 0.2–0.4 μm wide SEM; margo well-developed, widening towards pole and equator, mostly psilate, with few nanoechini/-verrucae SEM; colpus membrane not observed (Figs. 1C, 3A, 3G; Plate S03 in File S3).

Locality

Miller Clay Pit, Henry County, Tennessee, USA (Table 1).

Remarks

Pollen grains of this morphotype show the exclusive morphology of pollen of two of the three extant Tripodanthus species: Tripodanthus acutifolius (Ruiz & Pav.) Tiegh. and Tripodanthus flagellaris Tiegh. as described and figured by Feuer & Kuijt (1980) and Grímsson, Grimm & Zetter (2017). The recently described Tripodanthus belmirensis F. J. Roldán & Kuijt has a different, more compact type of pollen (Roldán & Kuijt, 2005). All species are endemic to South America (e.g. Amico et al., 2012).

Use as age constraint

Representing a characteristic pollen type known only from two modern species of the same genus, Miller Clay Pit MT2, MT3, and the Aamaruutissaa MT could be used as minimum age constraints for the MRCA of Tripodanthus with respect to Tripodanthus belmirensis and its different pollen. We followed a more conservative approach here. Tripodanthus is often reconstructed as the first diverging branch within the Psittacanthinae, followed in most trees by Psittacanthus. The latter is a genus with diverse pollen (Feuer & Kuijt, 1979), including morphologies more similar to those of Tripodanthus and its fossil counterparts than to the remainder of the subtribe (and Tripodanthus belmirensis). The remainder is characterised by compact B-type pollen with minute to indistinct sculpturing and pollen grains of the Type C (Passovia pyrifolia, Dendropemon) and D (Oryctanthus). Compact B-type, C-type and D-type pollen occur much later in the fossil record (File S4) and are completely missing in our samples. The latter three types appear to be derived. Taking all evidence into account, we cannot exclude the possibility that Tripodanthus acutifolius and Tripodanthus flagellaris simply retained a more ancestral pollen type of the Psittacanthinae. The fossil pollen grains hence would not indicate the presence of the genus Tripodanthus in North America and Greenland, but of extinct, northern-hemispheric or ancestral members of the Psittacanthinae, thereby informing a conservative minimum age for the MRCA of Psittacanthinae and their sister clade. Unfortunately, this sister clade, if not constrained (scenario 4), is not resolved with meaningful support. As a trade-off, we used the Aamaruutissaa MT—the most precisely dated pollen of the Tripodanthus-like MTs and likely younger than their American counterparts—as minimum age constraint for the MRCA of the Psittacanthinae lineage for the rooting scenarios 1–3 (under the assumption that crown radiation within the Psittacanthinae must have started before the time of a loranth that produced Tripodanthus-like pollen and thrived in Greenland, far outside the modern distribution area of the family.)

Miller Clay Pit MT3, aff. Tripodanthus

Description

Pollen, oblate, slightly concave-triangular in polar view, no undistorted equatorial view available, equatorial apices truncated; pollen small, equatorial diameter 20.0–21.7 μm in LM, 19.6–21.3 μm in SEM; syn(3)colpate, colpi narrow; exine 0.9–1.6 μm thick, nexine thinner than sexine, intercolpial nexine thickening at pole, sexine thickened in area of mesocolpium (LM); tectate; sculpturing psilate in LM, (micro)baculate and perforate in area of mesocolpium in SEM, (micro)bacula densely packet, (micro)bacula 0.4–1.8 μm long, 0.1–0.2 μm wide; margo well developed, markedly broader in equatorial regions, margo faintly nano- to micro-rugulate SEM; colpus membrane nanoverrucate to granulate SEM (Figs. 1D, 3B, 3C, 3H, 3I; Plate S04 in File S3).

Locality

Miller Clay Pit, Henry County, Tennessee, USA (Table 1).

Remarks

General outline and size of the Miller Clay Pit MT3 is very similar to those of Miller Clay Pit MT2. The main difference is that the margo in Miller Clay Pit MT3 can be faintly rugulate, a feature not observed in Miller Clay Pit MT2 or the two extant species with nearly identical pollen. Also, the mesocolpium is perforate in Miller Clay Pit MT3; a feature not seen in Miller Clay Pit MT2 or extant Tripodanthus. As a trend, the sculptural elements are narrower and can be much longer than in Miller Clay Pit MT2 pollen.

Use as age constraint

See Miller Clay Pit MT2.

Aamaruutissaa MT, aff. Tripodanthus

Description

Pollen, oblate, slightly concave-triangular in polar view, no undistorted equatorial view available, equatorial apices truncated; pollen small, equatorial diameter 18.6–22.0 μm in LM, 18.5–21.5 μm in SEM; syn(3)colpate; exine 1.0–1.3 μm thick, nexine thinner than sexine, intercolpial nexine thickening at pole (LM); tectate; sculpturing psilate in LM, nano- to microbaculate in area of mesocolpium in SEM, bacula 0.3–1.1 μm long, 0.2–0.3 μm wide SEM; margo well developed, margo faintly nano- to microrugulate SEM; colpus membrane nanoverrucate to granulate SEM (Figs. 1E, 3D, 3J; Plate S05 in File S3).

Locality

Aamaruutissaa, southeast Qeqertarsuatsiaq Island, western Greenland (Table 1).

Remarks

This pollen type has previously been figured as Loranthaceae gen. et spec. indet. (Manchester, Grímsson & Zetter, 2015: Figs. 2A–2C). Like Miller Clay Pit MT2 and MT3, it is nearly indistinguishable from the pollen of the two original species of Tripodanthus, Tripodanthus acutifolius and Tripodanthus flagellaris. The Aamaruutissaa MT pollen combines the mesocolpial sculpturing seen in Miller Clay Pit MT2 with the shape and margo seen in Miller Clay Pit MT3. With respect to the modern species, both the Tennessee (Miller Clay Pit MT2, MT3) and Greenland pollen grains (Aamaruutissaa MT) were possibly produced by the same genus or at least closely related taxa of the same loranth lineage (Psittacanthinae).

Use as age constraint

See Miller Clay Pit MT2.

Stolzenbach MT, pollen of ambiguous affinity

Description

Pollen, oblate, trilobate in polar view, no undistorted equatorial view available, equatorial apices obcordate, interapertural areas (mesocolpia) sunken; pollen small, equatorial diameter 12.1–15.4 μm in LM, 11.7–15.3 μm in SEM; syn(3)colpate, colpi narrow; exine 0.7–0.9 μm thick, nexine thinner than sexine; tectate; sculpturing psilate in LM, microechinate in area of mesocolpium in SEM, echini stout with blunt apices, 0.4–0.8 μm long, 0.3–0.8 μm wide at base SEM; margo well developed, broad, covering the grain’s surface in polar view, microrugulate, granulate SEM; colpus membrane mostly granulate SEM (Figs. 1F, 2B, 2I; Plate S06 in File S3).

Locality

Stolzenbach underground coalmine, Kassel, Germany (Table 1).

Remarks

Size, outline, and form of the pollen, and SEM sculpturing in the area of the mesocolpium is most similar to what has been observed in pollen of modern monotypic root-parasites Nuytsia and Gaiadendron, and the Lorantheae Muellerina (Ileostylinae). Despite this general similarity, the pollen differs from the modern ones and pollen with affinity to Nuytsia reported from the Miller Clay Pit, Tennessee (Miller Clay Pit MT1), visually (compare overviews in Figs. 2B, 2E, 2F, 2G) and regarding its sculpturing. The Stolzenbach MT echini are sparsely packed and broader at the base; the striae on the margo are flatter and broader. The pollen may well represent an unrelated, extinct lineage or ancestral taxon with affinities to both the root-parasitic lineages and/or the Lorantheae.

Use as age constraint

Although the pollen cannot be assigned to any modern genus or lineage, it is an early Central European representative of the common Pollen Type B of Loranthaceae. Its morphology is in many aspects primitive within the (B-type) Loranthaceae, hence, the similarity with Nuytsia/Miller Clay Pit MT1, Gaiadendron, and Muellerina (the only Lorantheae known so far with a striate ornamentation). Its morphology, place, and age would fit for an early precursor or extinct sister lineage of the Lorantheae. Taken together with the coeval pollen from North America and Greenland, it provides evidence for the onset of diversification of B-type pollen lineages including the possible establishment of the Lorantheae. Hence, it was used to constrain the minimum age of the MRCA of all Loranthaceae (rooting scenario 2; clock-based root) or Loranthaceae with B-type pollen (rooting scenario 3; pollen morphology-informed root).

Profen MT1, pollen of unknown affinity

Description

Pollen, oblate, trilobate in polar view, elliptic in equatorial view, lobes very narrow, equatorial apices obcordate, interapertural areas (mesocolpia) sunken; pollen small, polar axis 10.0–12.3 μm long in LM, 9.5–11.0 μm long in SEM, equatorial diameter 13.8–17.5 μm in LM, 11.9–13.8 μm in SEM; syn(3)colpate; exine 0.9–1.1 μm thick, nexine thinner than sexine; tectate; sculpturing psilate in LM, nanoechinate, nanobaculate, granulate in area of mesocolpium in SEM, echini/bacula 0.3–0.6 mm long, 0.2–0.4 μm wide SEM; margo well-developed, covering nearly the entire surface of the grain in polar view, faintly microrugulate SEM; colpus membrane nanoverrucate to granulate SEM (Figs. 1G, 1H, 2C, 2J; Plate S07 in File S3).

Locality

Profen, Leipzig, Central Germany (Table 1).

Remarks

Like the Stolzenbach MT pollen this fossil pollen type has no direct modern counterpart. These small, narrow-lobate pollen grains with their finely sculptured, deeply sunken mesocolpia characteristic of the Profen MT1 pollen are not found in any modern taxon, but bear some similarity to the younger (Oligocene) pollen of Theiss (see below). Equally small pollen grains are only known from the root-parasites Nuytsia and Gaiadendron, and the Lorantheae Muellerina. Equally minute sculpturing is only found in otherwise completely different, and putatively derived pollen of deeply nested (phylogenetically) Psittacanthinae and Lorantheae.

Use as age constraint

Showing a unique combination of putatively primitive and derived morphological features, this pollen could only be used to constrain the minimum age of the MRCA of all Loranthaceae with B-type pollen.

Profen MT2, aff. Notanthera

Description

Pollen, oblate, straight- to slightly concave-triangular in polar view, no undistorted equatorial view available, equatorial apices obcordate; pollen small, equatorial diameter 21.5–23.1 μm in LM, 18.3–19.6 μm in SEM; syn(3)colpate, colpi narrow; exine 1.1–1.4 μm thick, nexine thinner than sexine, intercolpial nexine thickening at pole, sexine thickened in area of mesocolpium (LM); tectate; sculpturing psilate in LM, nano-echinate/-baculate, perforate in area of mesocolpium in SEM, echini/bacula stout, sometimes fused, 0.2–0.4 μm long, 0.2–0.4 μm wide SEM; margo well-developed, slightly widening towards pole and equator, psilate to faintly microrugulate SEM; colpus membrane nanoverrucate to granulate SEM (Figs. 1I, 1J, 4A, 4B, 5A, 5B; Plate S08 in File S3).

Locality

Profen, Leipzig, Central Germany (Table 1).

Remarks

Form and sculpturing of pollen grains of this morphotype are remarkably similar to those of Notanthera heterophylla (Feuer & Kuijt, 1980: Fig. 5). Notanthera heterophylla is one of two species that comprise the two monotypic genera of the South American Notantherinae; a subtribe of the Psittacantheae neither resolved as clade nor rejected with high support in molecular-phylogenetic inferences (Grímsson, Grimm & Zetter, 2017: Files S1, S6). The sculpturing of Profen MT2 is furthermore in line with the description and TEM image provided by Feuer & Kuijt (1980).

Systematic note

The second species included in the Notantherinae, Desmaria mutabilis (Poepp. & Endl.) Tiegh. ex B.D.Jacks., has not only a different pollen (Feuer & Kuijt, 1980; Grímsson, Grimm & Zetter, 2017) but is also genetically distinct (Fig. 7).

Figure 7 Plastid and nuclear species trees for the complete taxon set.

No high-supported conflict is found; both datasets recognise the same main clades, while failing to resolve most of the deeper inter-clade relationships. Particularly, the phylogenetic position of tribes/subtribes with few, often monotypic, genera (root parasitic Nuytsieae, Gaiadendreae, aerial parasitic Ligarinae, Notantherinae, and Tupeinae) is essentially unresolved. Local differences in the topologies and odd placements are often related to species with large amount of missing data. Stippled terminal branches have been reduced by factor 2.

Use as age constraint

This pollen can inform the minimum root age for the lineage leading to Notanthera, i.e. the minimum age of the MRCA of Notanthera and Elytrantheae (scenarios 1–3; preferred topology based on the taxon-reduced data set) or Notanthera and Psittacanthinae (scenario 4; topology constrained to fit with Su et al., 2015: Fig. 1B).

Profen MT3, pollen of the Elytrantheae clade

Description

Pollen, oblate, convex-triangular in polar view, no undistorted equatorial view available, equatorial apices more or less truncated; pollen small, equatorial diameter 20.0–21.5 μm in LM, 19.2–20.0 μm in SEM; syn(3)colpate, colpi very narrow at equatorial apices, widening towards the pole; exine 0.9–1.1 μm thick, nexine thinner than sexine (LM); tectate; sculpturing psilate in LM, mostly nano-baculate to -echinate in area of mesocolpium in SEM, bacula/echini 0.2–0.5 μm long, 0.1–0.2 mm wide SEM; margo well developed, covering the equatorial apices, mostly psilate, with few nano-bacula/-echini in polar area, forming triangular protrusions at pole SEM; colpus membrane nano-echinate/-verrucate to granulate SEM (Figs. 1K, 4C, 5C; Plate S09 in File S3).

Locality

Profen, Leipzig, Central Germany (Table 1).

Remarks

The combination of characters (syncolpate with widening colpi, margo with triangular protrusions and sculpturing reminiscent of the mesocolpium in polar area, sculpturing of mesocolpium nano-baculate/-echinate) is today only found in members of the Elytrantheae. With respect to studied modern Elytrantheae, the pollen of Profen MT3 resembles the most that of Peraxilla tetrapetala (L.f.) Tiegh. (Fig. 4), but the sculpturing elements are more slender and higher (Fig. 5). The sculpturing in the mesocolpium (dimension and density of sculptural elements) matches grains included in another morphotype found at Profen (Profen MT4; Fig. 5).

Use as age constraint

Here we used Profen MT3, MT4 and MT5 to constrain the root age of the Elytrantheae, i.e. the minimum age of the MRCA of Notanthera and Elytrantheae (scenarios 1–3). Further studies of modern pollen of Elytrantheae at and below the genus level and more genetic data are needed to decide whether the Profen MT3, and the related Profen MT4 and MT5, are already indicative for a first divergence within the Elytrantheae and can be placed more decisively within the Elytrantheae subtree.

Profen MT4, possible pollen of the Elytrantheae clade

Description

Pollen, oblate, concave-triangular to trilobate in polar view, no undistorted equatorial view available, equatorial apices T-shaped; pollen small, polar axis 11.3–15.0 μm long in LM, equatorial diameter 17.5–25.0 μm in LM, 14.3–20.0 μm in SEM; demisyn(3)colpate, colpi short SEM, widening towards the pole forming a polar depression (polar sexine reduced); exine 1.1–1.3 μm thick, nexine thinner than sexine, nexine hexagonally thickened in polar area (LM); tectate; sculpturing psilate in LM, mostly nano-baculate/-echinate in SEM, bacula/echini 0.3–1.1 μm long, 0.1–0.4 μm wide at base SEM; margo indistinct in polar area, more prominent in equatorial regions, faintly microrugulate, covered by nano-bacula/-echini in polar area SEM; colpus membrane nano-echinate/-verrucate to granulate SEM (Figs. 1L–1O, 4D, 4E, 5D; Plate S10, S11 in File S3).

Locality

Profen, Leipzig, Central Germany (Table 1).

Remarks

This pollen type has previously been figured (Manchester, Grímsson & Zetter, 2015: Figs. 2D–2F), designated as “Loranthaceae gen. et spec. indet.” Sculpturing of Profen MT4 is somewhat variable; dimensions, density and shape of sculptural elements resemble those in Profen MT3 and Profen MT5 (see later), or are overlapping between both. Regarding its form (trilobate with T-shaped equatorial apices) and lacking a distinct margo in the polar area, the pollen differs from all modern members of the Elytrantheae. In this aspect, it is similar to the pollen of Ligaria (Psittacantheae: Ligarinae), a genus with ambiguous phylogenetic affinities to other New World genera (Grímsson, Grimm & Zetter, 2017: File S1, Figs. S6-1–9). Also in Ligaria, the sexine is reduced in the polar area (Fig. 4), the generally very narrow colpi are fusing in a triangular polar depression (a feature only seen in Ligaria and its putative relative Tristerix). Ligaria pollen grains are furthermore distinctly microbaculate (Fig. 5). Bacula are found in all three Profen morphotypes linked to the Elytrantheae lineage, but are rare or absent in the modern members of this clade.

Use as age constraint

See Profen MT3.

Profen MT5, probable pollen of the Elytrantheae clade

Description

Pollen, oblate, straight-triangular in polar view, elliptic to subrhombic in equatorial view, equatorial apices broadly rounded; pollen small to medium, polar axis 7.5–11.5 μm long in LM, equatorial diameter 21.5–30.0 μm in LM, 18.7–24.4 μm in SEM; demisyn(3)colpate, widening towards pole, terminating halfway between pole and equator SEM; exine 1.1–1.3 μm thick, nexine thinner than sexine (LM); tectate; sculpturing psilate in LM, mostly nano- to micro-baculate/-echinate in area of mesocolpium in SEM, bacula/echini 0.3–0.7 μm long, 0.1–0.3 mm wide at base; margo distinct but not raised, mostly psilate, with few nano-bacula/-echini in polar area, forming triangular protrusions at pole SEM; colpus membrane nano-echinate/-verrucate to granulate SEM (Figs. 1P, 1Q, 4F, 5E; Plate S12 in File S3).

Locality

Profen, Leipzig, Central Germany (Table 1).

Remarks

The pollen fits with the morphotypes seen in modern members of the Elytrantheae, although its combination of characters is unique. Small, demisyncolpate, (sub)rhombic pollen grains are (so far) only known from Amylotheca, which differ from the fossil pollen by their outline in polar view (Fig. 4) and sculpturing (Fig. 5). Regarding the latter, Profen MT5 is very similar to grains included in Profen MT3. Both, Profen MT3 and MT5, differ from the third morphotype with possible affinities to Elytrantheae (Profen MT4) by their demisyncolpate grains (Fig. 4). Regarding the mesocolpium, Profen MT5 shows the densest sculptured mesocolpium of all three morphotypes (Fig. 5).

Use as age constraint

See Profen MT3.

Changchang MT, aff. Amyeminae vel Scurrulinae

Description

Pollen, oblate, concave-triangular to broadly trilobate in polar view, no undistorted equatorial view available, equatorial apices broadly rounded; pollen small, equatorial diameter 21.1–24.4 μm in LM, 19.1–21.8 μm in SEM; syn(3)colpate; exine 0.9–1.1 μm thick, nexine thinner than sexine; tectate; sculpturing psilate in LM, nanoverrucate to granulate, perforate in SEM, granula partly fused; margo well developed, psilate, widening towards the equator, usually covering the entire apex SEM; colpus membrane granulate; rhombic structures (opercula) covering equatorial apices SEM (Figs. 1R, 1S, 6A, 6B, 6H, 6I; Plate S13 in File S3).

Locality

Changchang Basin, Jiazi Town, northern Hainan Island (Table 1).

Remarks

The minute sculpturing and its basic form link this pollen to the Lorantheae, in particular to the Scurrulinae Taxillus and Scurrula (unresolved within Clade J) on one hand, and Amyema (Amyeminae; Clade I in Vidal-Russell & Nickrent, 2008a, sister clade of Clade J) on the other hand. The pollen could be described as a Scurrulinae pollen with an Amyema-like margo. A unique feature not found in any modern Loranthaceae so far are the operculum-like triangular structures of the equatorial apices. Pollen of the two first diverging, long-branched lineages in Lorantheae (Clades G and H in Vidal-Russell & Nickrent, 2008a; cf. Su et al., 2015: Figs. 1B, S7; Grímsson, Grimm & Zetter, 2017: Fig. 2) are markedly distinct. Thus, we think that this pollen belongs to an extinct or ancestral Lorantheae lineage related to the core Lorantheae (= Clades I and J according Vidal-Russell & Nickrent).

Use as age constraint

Based on its morphology, the Changchang MT could already represent an early member of the Lorantheae core clade, i.e. would inform a minimum age of the MRCA of Lorantheae core clade and its sisterclade Ileostylinae. However, some molecular data sets indicate a sister relationship between Ileostylinae and Loranthinae (cf. Grímsson, Grimm & Zetter, 2017: Files S1, S6). Furthermore, more information on pollen morphology in Lorantheae would be needed to exclude the possibility that the Changchang MT is correctly recognised as a representative of the Lorantheae core clade. Two of the four genera of the sister lineages of the core Lorantheae (Loranthinae, Ileostylinae) have not yet been studied palynologically and little is known on the other Amyeminae genera, the first diverging branch within the core Lorantheae. Hence, we opted for a more conservative approach and used the Changchang MT to constrain the MRCA of all Lorantheae.

Theiss MT

Description

Pollen, oblate, trilobate in polar view, emarginate in equatorial view, lobes very narrow, equatorial apices rounded; pollen small, polar axis 8.3–11.7 mm long in LM, 6.5–8.3 μm long in SEM, equatorial diameter 10.0–15.0 μm in LM, 10.3–11.8 μm in SEM; demisyn(3)colpate; exine 0.9–1.2 μm thick, nexine thinner than sexine (LM); tectate; sculpturing psilate in LM, nano- to microverrucate in area of mesocolpium in SEM, verrucae often fused, widely spaced, verrucae composed of conglomerate granula SEM; margo well-developed, covering nearly the entire surface of the grain in polar view, faintly microrugulate, granulate SEM; colpus membrane unknown (Figs. 1T–1X, 2D; Plate S14, S15 in File S3).

Locality

Theiss, borehole southeast of Krems, Lower Austria (Table 1).

Remarks

This fossil pollen type has no direct modern counterpart. A unique feature is the widely spaced verrucae in area of mesocolpium. Demisyncolpate grains evolved at least three times in the Loranthaceae: in Amylotheca (Elytrantheae), in the Cladocolea-Struthanthus lineage and Passovia (Psittacanthinae), and Tapinanthus (T. bangwensis [Engl. & K.Krause] Danser, T. ogowensis [Engl.] Danser; Lorantheae core clade). The fossil pollen shares no other features with either Elytrantheae or Psittacanthinae. Grains with narrow (deflated) equatorial lobes, in which the margo extends beyond the mesocolpial plane, are so far only known from several members of the Lorantheae core clade (e.g. Englerina, Globimetula, Phragmanthera). Mesocolpia with exclusively nanoverrucate to granulate sculpturing are only found in members of the Lorantheae. For instance, the Tapinanthinae (core Lorantheae) Actinanthella has emarginate, trilobate (to convex-triangular) pollen grains with a well-developed, mostly psilate margo and nanoverrucate to granulate mesocolpium, but they differ from the fossil pollen by their size and zonocolpate apertures.

Use as age constraint

Due to the unique morphology, yet superficial knowledge about pollen evolution in Lorantheae (see Changchang MT), we decided against using the Theiss MT to constrain a node higher up in the tree (e.g. the MRCA of Tapinanthinae and Emelianthinae).

Altmittweida MT, aff. Helixanthera

Description

Pollen, oblate, convex-triangular in polar view, emarginate in polar view, equatorial apices broadly obcordate; pollen small, polar axis 4.4–5.5 μm long in LM, equatorial diameter 14.4–17.8 μm in LM, 13.7–16.0 μm in SEM; syn(3)colpate; exine 0.9–1.1 μm thick, nexine thinner than sexine, intercolpial nexine thickening at pole, sexine partly reduced in polar area SEM; tectate; sculpturing psilate in LM, nano- to micro-verrucate, granulate in SEM, verrucae composed of conglomerate granula (Fig. 6J); margo psilate to microverrucate, granulate; colpus membrane nanoverrucate to granulate SEM (Figs. 1Y–1Ä, 6C, 6D, 6J; Plate S16 in File S3).

Locality

Altmittweida, Saxony, Germany (Table 1).

Remarks

This pollen type has previously been figured by Kmenta (2011: Plate 11, Figs. 1–3) as “Loranthaceae gen. et spec. indet.” Pollen very similar to this fossil pollen can be found in two extant species of the Lorantheae: Amyema gibberula Danser (type genus of Amyeminae, Clade I according Vidal-Russell & Nickrent, 2008a) and Helixanthera kirkii (Oliv.) Danser (Grímsson, Grimm & Zetter, 2017). Both species are similar in outline (convex-triangular, emarginate) and sculpturing (margo indistinct, with similar sculpturing than adjacent mesocolpium). In LM, Amyema shows a distinct hexagonal thickening of the polar nexine, whereas in Helixanthera the thickening covers a larger area of the grain and is most dominant in the intercolpial areas; the latter can be seen in the fossil pollen. The flanks of the equatorial apices in the equatorial plain are straight in Helixanthera and the fossil, whereas they are continuously curved in Amyema. In addition, the polar depression in Helixanthera and the fossil are identical in all details in SEM (Figs. 6C, 6D, 6F), whereas in Amyema the polar margo is more distinct and shows three small triangular protrusions (Fig. 6E).

Use as age constraint

The phylogenetic position of Helixanthera within the core Lorantheae is uncertain. Nucleotide data has been produced for three species including Helixanthera kirkii, but the data are partly problematic and too fragmentary. Helixanthera kirkii (only nuclear data available, only species palynologically studied so far) nests deep within the Lorantheae core clade, and H. parasitica Lour. (only plastid data available) is sequentially divergent from all Lorantheae and effectively unplaced (see Fig. 7 and File S1 for further details). The third and best covered species, H. coccinea Danser, groups with species of Dendrophthoe in agreement with the current systematic scheme, but its pollen is yet to be studied. Thus, Helixanthera has not been included in the taxon-reduced species-consensus dataset used here for the molecular dating. A conservative use could be constraining the minimum age of the MRCA of the Clade J (i.e. Scurrulinae, Dendrophthoinae, Emelianthinae, and Tapinanthinae; see Discussion).

Inferences

Basic signal in the harvested molecular data

Our inferences, based on species-consensus sequences and different sets of data (see File S1 and files included in OSA), did not reveal any well-supported conflict between the nuclear and plastid gene regions (Fig. 7). Inclusion or exclusion of the most divergent, length-polymorphic non-coding (plastid) trnL/LF region showed little effect on the optimised ML topologies and BS support values. When not including any long-branching outgroups, the data largely fails to group the root parasitic taxa. Hence, there is a lack of support for a root parasitic grade. An according subtree (e.g. Su et al., 2015: Fig. 1B) draws its support exclusively from the matK gene data and is enforced if long-branching sistergroups of the Loranthaceae are included (Grímsson, Grimm & Zetter, 2017: File S6). Overall, the single- and oligo-gene species-consensus trees showed the same principal topology as earlier found using genus-consensus sequences (Grímsson, Grimm & Zetter, 2017: Figs. 2, 3). However, species of the same genus were not necessarily reconstructed as siblings. In the case of Helixanthera, Psittacanthus (nuclear and plastid data), and Plicosepalus (plastid data only), the branches separating the putative siblings received no high support, while the opposite was true for Amyema, Tapinanthus (nuclear and plastid data), Amylotheca, Lepidaria, and Oncocalyx (plastid data only). In terms of genetic-phylogenetic distances, the species of Helixanthera show the least coherence at the genus level. Aside from this, several clades were consistently reconstructed and usually received moderately high to unambiguous support (BS ≥ 70) from different data sets (Fig. 7): (1–4) the Old World Lorantheae with three subclades (Loranthinae, Ileostylinae, core Lorantheae), (5, 6) the Amyeminae (except Baratranthus axanthus) and Scurrulinae within the core Lorantheae; (7) the New World Psittacanthinae (except for Aetanthus, which is poorly sampled in our data set); and (8) the Elytrantheae (poorly supported based on nuclear data due to faint discriminating signals). The positions of the other mostly monotypic genera of the family remained unresolved; alternative splits representing deep relationships generally received low support. Bayesian inference is more decisive, with PP ∼ 1.00 found for all major splits with moderately high to high BS support (BSML ≥ 77) and several splits with low BS support (see Materials and Methods). Some of the deepest splits that received BSML < 35, received PP > 0.5 (all alternatives with PP < 0.2). A split between root and aerial parasites is not supported by any analysis with BSML/PP > 20/0.2. A detailed account regarding topological ambiguity of inferences using the currently available molecular data can be found in File S5.

The divergence in the covered gene regions is substantial (see branch-lengths in Fig. 7); the resulting terminal ‘noise’ appears to obscure any signal that may allow for the discrimination of deeper phylogenetic splits. This may explain to some degree, in addition to the relatively high proportion of missing data, the low resolution capacity of comprehensive species-level data sets. When the taxon set was reduced to only those species with full data coverage, support along the backbone and towards the leaves of the Loranthaceae tree increased. This reduction also showed a positive effect on the dating: using the complete taxon set and matrices with numerous data gaps, ESS values converged very slowly (rooting scenarios 1 and 3) or not at all (rooting scenario 2; see also File S1).

Alternative clock-based roots

For four of the five comprehensive datasets (all taxa, different sets of gene samples), the clock-inferred root was placed between the predominately Old World Lorantheae and a mostly southern hemispheric, American-Australasian clade collecting all three root parasitic genera and the members of the other two aerial parasitic tribes, the (probably paraphyletic) Psittacantheae and (putatively monophyletic) Elytrantheae (Table 2). In the case of the most-inclusive data set (all taxa, all gene regions), the root was shifted by two nodes and placed within the Lorantheae subtree, splitting the genetically divergent subtribes Loranthinae and Ileostylinae from the remainder of the Lorantheae (= Clade J according to Vidal-Russell & Nickrent, 2008a). The subsequent evolutionary scenario would imply that root parasites and other southern hemispheric lineages share an ancestor with only the Loranthinae and Ileostylinae. This would mean a paraphyletic Lorantheae tribe, which is highly unlikely (Nickrent et al., 2010; Su et al., 2015; Grímsson, Grimm & Zetter, 2017). Thus, this alternative root was not further considered. In contrast to these roots, the taxon-reduced, less ‘gappy’ dataset (42 species covering, at least partly, all included gene regions) recovered the outgroup-inferred root, with Nuytsia as sister to all other loranths.

Temporal framework for pollen evolution in Loranthaceae

Following our clock-rooting results and those of earlier studies, we applied three different root constraints to judge potential effects of topological uncertainties regarding the primary relationships on the dating estimates (Figs. 8–9; Table 3). In addition, we constrained our data to the topology of the Loranthaceae subtree as shown in Su et al. (2015; scenario 4), which—according to an expert on the group—is the most correct one to date (but see Grímsson, Grimm & Zetter, 2017: File S6). We find that independent of the position of the root and exact structure of the backbone topology, primary divergences in Loranthaceae were terminated by the end of the Eocene at the latest (Table 3). The posterior estimates of the evolutionary rates per gene were equivalent in all rooting scenarios (Table 4) and slightly higher for the fourth scenario in which the topology was constrained to the one of Su et al. (2015). The estimated rates for matK and trnL/LF were within the range of mean rates reported for coding and non-coding plastid gene regions (7 × 10−5–8 × 10−3 substitutions per million years; e.g. Wolfe, Li & Sharp, 1987; Palmer, 1991; Chen et al., 2012; see also Guzmán & Vargas, 2010; Désamoré et al., 2011; Lockwood et al., 2013). The robustness of our estimations are further supported by the fact that the observed phases of increased diversification (number of coexisting lineages) and stagnation concur with key events in Cenozoic climate and vegetation evolution (Fig. 8). Most crown group radiation, the formation of the modern genera, apparently happened no later than the Miocene. Based on the limited species coverage, it is impossible to estimate when intra-generic radiation stepped in, and at which point closer related genera became isolated and diverged.

Figure 8 Lineage-through-time plots for Loranthaceae as inferred based on three different rooting scenarios or enforcing the topology of Su et al. (2015; scenario 4).

Background shows the stable-isotope-based (marine sediments) global temperature curve with main climatic events annotated at the bottom (after Zachos et al., 2001). Increased diversification of Loranthaceae is inferred for time-scales when the global mean temperature was at least ∼5° C higher than today (middle to late Eocene; late Oligocene to mid-Miocene).

Figure 9 A dated phylogeny of Loranthaceae using the pollen-informed root (rooting scenario 3).

The chronogram is based on a concatenated data set including two nuclear ribosomal RNA genes (18S and 25S rDNA), two coding plastid genes (rbcL, matK) and the trnL/LF region. The taxon set has been reduced to species with sufficient data, i.e. data covering all included gene regions. Node heights (divergence ages) are medians, grey bars indicate the 95%-highest-posterior-density intervals; labels at branches indicate posterior probabilities for those branches that did not receive unambiguous support. Triangular doodles represent pollen used as age priors for the according nodes: green—Central Europe; red—North America (including Greenland); yellow—East Asia. ECO, Eocene warm phase; MCO, Miocene warm phase (see Fig. 8).

Table 3 Results of the dating analyses using the reduced taxon data set and different rooting scenarios.

Node	Rooting scenario 1	Rooting scenario 2	Rooting scenario 3	Scenario 4	Av. Medians	Abs. min	Corresponds to	
L.b.	Median	U.b.	L.b.	Median	U.b.	L.b.	Median	U.b.	L.b.	Median	U.b.	
Loranthaceae crown	52.6	50.1	47.8	51.5	49.1	46.9	56.1	50.8	47.3	48.0	45.4	43.0	48.9	43.0	Earliest	Lutetian	
Nuytsia root	52.6	50.1	47.8	50.4	48.1	45.9	47.4	41.6	34.2	48.0	45.4	43.0	46.3	34.2	Latest	Priabonian	
Atkinsonia root	46.8	43.8	40.7	45.7	43.1	40.5	47.5	44.3	40.9	45.9	43.9	42.0	43.8	40.5	Early	Bartonian	
Gaiadendron root	46.5	43.7	40.7	45.7	43.1	40.6	47.4	44.3	41.1	45.1	43.2	41.5	43.6	40.6	Early	Bartonian	
Tristerix root	52.2	49.7	47.3	48.0	44.4	38.9	47.4	41.6	34.2	40.4	37.0	31.9	43.2	31.9	Latest	Priabonian	
Tupeia root	49.7	47.2	44.8	48.0	44.4	38.9	56.1	50.8	47.3	42.2	39.1	32.7	45.4	32.7	Late	Bartonian	
MRCA (aerial parasitic) new world taxa	52.2	49.7	47.3	48.7	46.8	44.9	50.8	48.5	46.2	42.8	41.4	40.2	46.6	40.2	Mid	Lutetian	
MRCA Desmaria–Ligaria	46.0	42.2	36.3	45.2	41.5	36.0	46.7	42.6	36.9	42.8	41.4	40.2	41.9	36.0	Mid	Priabonian	
Notanthera + Elytrantheae root*	46.8	43.8	40.7	45.7	43.1	40.5	47.5	44.3	40.9	[N/A]	43.7	40.5	Early	Bartonian	
MRCA Notanthera + Elytrantheae*	41.0	39.5	38.0	40.9	39.4	37.8	41.0	39.5	38.0	44.1	42.5	41.1	40.2	37.8	Latest	Bartonian	
Notanthera + Psittacanthinae root*	[N/A]	[N/A]	[N/A]	42.2	40.9	39.7	40.9	39.7	Early	Bartonian	
MRCA Notanthera + Psittacanthinae*	48.6	46.4	44.3	47.4	45.5	43.8	49.6	47.2	44.9	41.5	40.5	39.5	44.9	39.5	Early	Bartonian	
Psittacanthinae root	47.1	45.0	43.0	46.2	44.4	42.6	47.9	45.5	43.4	41.5	40.5	39.5	43.8	39.5	Latest	Lutetian	
Psittacanthinae crown	41.4	40.4	39.5	41.3	40.3	39.4	41.5	40.6	39.6	29.8	22.8	16.7	36.0	16.7	Mid	Bartonian	
Elytrantheae root	41.0	39.5	38.0	40.9	39.4	37.8	41.0	39.5	38.0	42.6	41.2	39.6	39.9	37.8	Latest	Bartonian	
Elytrantheae crown	38.5	33.4	26.7	38.2	33.1	26.6	38.5	33.5	27.0	35.1	27.2	20.2	31.8	20.2	Early	Chattian	
Lorantheae root	49.7	47.2	44.8	51.5	49.1	46.9	51.4	49.1	46.8	42.6	41.2	39.6	46.7	39.6	Mid	Lutetian	
Lorantheae crown	44.2	41.1	37.8	45.1	41.8	38.5	44.7	41.6	38.6	38.1	35.9	33.5	40.1	33.5	Earliest	Priabonian	
Core Lorantheae crown	35.2	31.2	27.0	35.9	31.6	27.4	35.6	31.7	27.4	29.8	26.5	22.9	30.2	22.9	Early	Chattian	
Notes:

Cells with same background colour refer to the same node. Medians closest to the arithmethic mean of all four scenarios (column ‘Av. Medians’) in bold, minimal age scenarios for each node (column ‘Abs. min’) highlighted by blue colour.

U.b., upper boundary; L.b., lower boundary, of the 95%-highest-posterior-density interval; MRCA, most recent common ancestor (can be inclusive or exclusive).

* If topology is unconstrained, Notanthera is placed as sister to Elytrantheae (BSML = 57; PP = 1.00); in Scenario 4, Notanthera is constrained as sister of Psittacanthinae (topological constraints derived from the tree shown in Su et al., 2015.

Table 4 Estimated substitution rates (per million years) for each of the used genetic markers.

Genetic marker	Rooting scenario 1	Rooting scenario 2	Rooting scenario 3	Scenario 4	
Median ucld.mean	
18S rDNA	2.5 × 10−4	2.5 × 10−4	2.5 × 10−4	2.9 × 10−4	
25S rDNA	6.5 × 10−4	6.6 × 10−4	6.3 × 10−4	8.2 × 10−4	
matK	10.1 × 10−4	10.1 × 10−4	10.0 × 10−4	11.7 × 10−4	
trnL/LFa	12.9 × 10−4	13.1 × 10−4	12.8 × 10−4	15.5 × 10−4	
Note:

Estimates are provided for all four tested topological hypotheses (rooting scenarios 1–3, and scenario 4 constraining the topology of Su et al. (2015)).

a Includes the trnL intron and downstream-located (5’) trnL exon (can be incomplete) and trnL–trnF spacer (complete).

Comparison of Bayes factors showed that rooting scenario 3, the pollen-informed root, is decisively superior (according Kass & Raftery, 1995) than the tested alternatives (Table 5). Thus, we chose rooting scenario 3 as the basis for our discussion and conclusion. The divergence between Tupeia (A-type pollen) and Loranthaceae with B-type pollen is placed in the early Eocene (∼50 Ma; Fig. 9, Table 3). A primary radiation followed shortly after (less than 2 myrs), and involved the formation of an essentially Old World (Lorantheae) and New World clade (root parasites, Elytrantheae, Psittacantheae). Subsequently, the first divergences in the New World clade occurred (≥43 Ma; Fig. 9). Crown group radiation in the Lorantheae started in the late Eocene (≥38 Ma) at the latest; the subclades and monotypic lineages (subtribes Psittacanthinae, Ligarinae, Notantherinae) of the probably paraphyletic Psittacantheae diverged at about the same time. A second major radiation phase took place ∼10 myrs later (latest in the Oligocene) and involved the Old World core Lorantheae (subtribes Amyeminae, Dendrophthoinae, Emelianthinae, Scurrulinae, Tapinanthinae) and New World Elytrantheae. Crown group radiation, the formation of lineages equalling most modern genera, commenced at about the same time and lasted till the mid-Miocene (≥9 Ma). In general, the genera root deeper, i.e. are older, in the (mostly) South American Psittacanthinae than in the Old World Lorantheae sublineages and the (mainly) Australasian Elytrantheae. Generic diversification culminates in the early to mid-Miocene, a time of ameliorated global climate (Zachos et al., 2001; see last section of Discussion).

Table 5 Ranking of the four tested topological configurations (three rooting scenarios, and scenario 4 constraining the topology of Su et al., 2015).

Rank	Scenario	Stepping-stone	Path-sampling	
MLE	BF	MLE	BF	
1	Rooting sc. 3
Tupeia sister to rest	−29457.1		−29456.0		
2	Rooting sc. 1
Nuytsia sister to rest	−29461.0	7.87	−29460.0	8.08	
3	Scenario 4
Su et al. (2015)	−29464.3	14.53	−29463.3	14.61	
4	Rooting sc. 2
(Lorantheae sister to rest)	−29466.3	18.54	−29465.7	19.43	
Note:

Ranking is based on marginal likelihood estimates (MLE) and Bayes factors (BF), calculated using two approaches, stepping-stone and path-sampling, implemented in beast (Baele et al., 2012, 2013).

Historical Biogeography

Pollen studied using SEM and subsequent node dating (Figs. 8, 9; Table 3) indicate that several major lineages of Loranthaceae were present in the Northern Hemisphere by the middle Eocene (Fig. 10A). The Eocene pollen record includes representatives of extinct or ancestral lineages with affinities to root-parasitic genera such as Nuytsia/Nuytsieae, but possibly also to the Lorantheae (Miller Clay Pit MT1, Stolzenbach MT, Profen MT1). In addition, today’s exclusively epiphytic lineages are present: Psittacanthinae in North America/Greenland (Miller Clay Pit MT2, MT3, Aamaruutissaa MT), Notanthera and Elytrantheae in Central Europe (Profen MT3, MT4 and MT5), and core group Lorantheae in East Asia (Changchang MT). All these records represent the earliest unequivocal fossil records of their respective groups. At least one of these lineages, the ancestral/extinct lineage bridging the root parasites and Lorantheae, persisted in Eurasia during the late Eocene and Oligocene (Theiss MT, Altmittweida MT; Fig. 10B) until today. These younger pollen types, which were not used as node age priors, are in good agreement with the dating estimates (Fig. 9). Furthermore, we noticed that none of the putatively derived pollen morphologies characteristic of certain members of the Psittacanthinae (compact B-type, C-type and D-type pollen) and Lorantheae (Loranthinae, Tapinanthinae-Emelianthinae; ± compact B-type pollen, B-type pollen with minute sculpturing, heteropolar grains) have been found (so far) in the older strata. Pollen records from the Miocene onwards, studied using LM and possibly representing a large range of Loranthaceae lineages with a B-type pollen, fall within the modern distribution area (Fig. 11), and potentially include such B types (File S4). The most derived C- and D-type pollen characteristic for Dendropemon, Passovia p.p. and Oryctanthus, which should be straightforwardly recognised with LM only, is rare and only known from late Miocene/sub-recent sedimentary rock formations. The dated trees predict an Oligocene/early Miocene age for the MRCA of Passovia pyrifolia and Oryctanthus (Fig. 9). If Loranthaceae with A-type pollen contributed to the pollen record of the family, they would not have been recognised as Loranthaceae, hence, are not included in our maps and File S4 (Figs. 10, 11).

Figure 10 Global distribution of Loranthaceae in the Paleogene, evidenced based on unequivocal palynological records.

(A) Eocene. (B) Oligocene. Maps are Mollweide views, projected through the prime meridian (Blakey, 2008; Global DVD © 2011 Colorado Plateau Geosystems Inc.).

Figure 11 Global distribution of Loranthaceae in the Neogene, evidenced based on unequivocal palynological records.

(A) Miocene. (B) Pliocene to recent. Asterisks indicate fossil occurrences; shaded/circum-lined areas in (B) reflect the modern-day distribution. Maps are Mollweide views, projected through the prime meridian (Blakey, 2008; Global DVD © 2011 Colorado Plateau Geosystems Inc.).

Well-resolved major clades of Loranthaceae are restricted to one or two adjacent biogeographic regions (Fig. 11). Except for Nuytsia/Nuytsieae (today only found in southwestern Australia), the fossil pollen records essentially reflect the modern situation, only extending the range of the respective New World and Old World lineages to higher latitudes of the Northern Hemisphere.

Discussion

Diagnostic value of Loranthaceae pollen for tracing modern lineages back in time

Pollen of various modern Loranthaceae have been studied using light (LM), transmission electron microscopy (TEM) and scanning-electron microscopy (SEM) (Feuer & Kuijt, 1978, 1979, 1980, 1985; Kuijt, 1988; Liu & Qiu, 1993; Han, Zhang & Hao, 2004; Roldán & Kuijt, 2005; Caires, 2012; Grímsson, Grimm & Zetter, 2017). In general, pollen of Loranthaceae—and other Santalales—reflect phylogenetic relationships and genetic-phylogenetic distances (Grímsson, Grimm & Zetter, 2017), which make them a valuable asset for biogeographic and dating studies. Some genera of putatively early diverging Loranthaceae lineages such as Nuytsia (monotypic Nuytsieae), Atkinsonia (bitypic Gaiadendreae, not resolved as clade in the molecular trees), the Psittacantheae Notanthera (bitypic Notantherinae), Ligaria and Tristerix (Ligarinae, not resolved as sibling genera), and Tripodanthus, Dendropemon, Orycanthus and Passovia p.p. (Psittacanthinae), show unique pollen types that have not been found in any other studied genus so far. Moreover, there is no indication that identical/highly similar pollen types evolved convergently in non-related Loranthaceae (or other Santalales). Non-unique pollen types are typically found in genera which are either part of the same, well-supported molecular clade (core Lorantheae; Elythrantheae; Psittacanthinae subclades), or shared with genera where the molecular data is indecisive regarding their exact phylogenetic position (Grímsson, Grimm & Zetter, 2017; this study).

Even though the modern situation makes it unlikely that—in the past—extinct lineages of Santalales or Loranthaceae have produced pollen mimicking those of modern, extant, but not closely related lineages, one needs to consider the possibility that a modern genus may have kept a more primitive (‘plesiomorphic’) pollen type of its evolutionary lineage. The Eocene and Oligocene pollen grains documented in this study show morphologies (1) not found in any modern taxon studied so far (Stolzenbach MT, Profen MT1, Theiss MT), or (2) found exclusively in a single modern genus (monotypic Nuytsia: Miller Clay Pit MT1, Tripodanthus with three extant species: Miller Clay Pit MT2, MT3, Aamaruutissaa MT; monotypic Notanthera: Profen MT2; phylogenetically problematic, see Fig. 7; Helixanthera: Altmittweida MT), or (3) are limited to a modern lineage (Elytrantheae: Profen MT3–5; core Lorantheae: Changchang MT) with none of the other modern species studied so far having an identical pollen. On the other hand, we found no pollen in our Eocene and Oligocene assemblages representing current-day diverse and widespread genera (such as Loranthus in Eurasia).

Extinct or ancestral pollen morphs of the Eocene and Oligocene of Europe

The shared pollen type of the South American root parasite Gaiadendron and the eastern Australian Lorantheae Muellerina (one of two genera in the subtribe Ileostylinae; the other has not been palynologically studied thus far) is a candidate for an ancestral, primitive and shared (‘symplesiomorphic’) morphology. The pollen of these two genetically and morphologically distinct modern genera are indistinct (Nickrent et al., 2010; Su et al., 2015: Fig. 2; Grímsson, Grimm & Zetter, 2017). The distinctly striate margo is a feature only seen in a few isolated, early diverging (Eocene) modern species/genera of ambiguous phylogenetic affinity (Fig. 9; Table 3). So far, no modern species showed an intermediate pollen type between the putatively plesiomorphic Gaiadendron–Muellerina pollen and the derived pollen characterising other members of the Lorantheae, e.g. the characteristically weakly oblate pollen of Loranthus. The Stolzenbach MT, Profen MT1, and Theiss MT of the Eocene and Oligocene of central Europe are equally small and share certain ornamental characteristics with the pollen of Gaiadendron–Muellerina such as a distinctly striate margo. Deviating features, e.g. more minute sculpturing of the mesocolpium, are shared with other members of the Lorantheae. This could make them candidates for an extinct lineage related to Lorantheae or ancestors of the Lorantheae subclades. At about the same time, more derived Lorantheae pollen grains can be found in the Eocene of East Asia (Changchang MT) and the Oligocene of Germany (Altmittweida MT), with clear affinities to the core Lorantheae. This provides conservative minimum estimates for the Lorantheae crown age, i.e. the divergence between Loranthinae, Ileostylinae, and core Lorantheae. Our dating estimates also indicate that there was a time gap of ca. 10 myrs between the formation and initial radiation of the Lorantheae and their subsequent diversification (Fig. 9; Table 3). Our current working hypothesis is that the Stolzenbach MT, Profen MT1, and Theiss MT, do in fact represent extinct sister lineages or precursors of the modern Old World Lorantheae (e.g. the Loranthinae). Whether these Loranthaceae extended into Africa or not, is unknown. The divergence between the East Asian Scurrulinae and the mostly African Tapinanthinae and Emelianthinae is placed in the Oligocene (Fig. 9), a time when substantial global cooling triggered the retreat of subtropical and tropical forests to low latitudes (Mai, 1995; Zachos et al., 2001). This event may have triggered the isolation between both clades and lead to the extinction of the ancestral pollen morphologies. Unfortunately, Africa is palaeo-palynologically understudied, so we do not know at which time the African Lorantheae with pollen grains typical for their modern members established. SEM studies of African palynofloras with Loranthaceae pollen from the Oligocene to Pliocene are desperately needed.

Pollen of Tripodanthus, a putative living palyno-fossil

Another case of a modern genus that conserved a primitive pollen morphology is evident from the Eocene pollen from North America and Greenland (Miller Clay Pit MT1, MT2; Aamaruutissaa MT). These pollen are highly similar to identical to pollen of two out of three species of the modern South American genus Tripodanthus; the third species has a more compact pollen somewhat similar to that of small-flowered species of the Psittacanthinae (Fig. S4; Feuer & Kuijt, 1985; Roldán & Kuijt, 2005; Amico et al., 2012; Grímsson, Grimm & Zetter, 2017). Tripodanthus is one of the earliest diverging Psittacanthinae (Figs. 7, 9; Vidal-Russell & Nickrent, 2008a; Grímsson, Grimm & Zetter, 2017). Pollen in the other represented genera of the Psittacanthinae (Passovia, Dendropemon, Struthanthus, Oryctanthus) appear strongly derived in comparison to that of Tripodanthus and part of Psittacanthus (Feuer & Kuijt, 1979, 1985), and include types that could be identified under LM. However, such pollen have not yet been reported from the fossil record except for the youngest strata (Bartlett & Barghoorn, 1973; Graham, 1990: File S4). Moreover, the current molecular data covers only a very limited fraction of the species in the Psittacanthinae, a clade palynologically well studied and diverse. So, at the moment, we lack a sound molecular framework to test hypotheses about pollen evolution within the group, and the group is genetically undersampled. Even so, our set of ML inferences highlights the shortcoming of the current generic concepts used for the group. So far, Tripodanthus is the only Psittacanthinae genus where the species/sequenced individuals show a relatively high topological coherence; an according, exclusive clade is supported by varying support (Fig. 7; Files S1, S5).

The Eocene Tripodanthus-like pollen of North America and Greenland might have been produced by extinct or ancestral members of the Psittacanthinae, rather than an ancient member of the Tripodanthus-lineage. It may merely confirm the existence of the New World Psittacanthinae clade in the Eocene of North America and Greenland, and should be linked with a deeper node. Using LM, Loranthaceae pollen (Gothanipollis sp.) has been recorded from North and South America from the early Eocene onwards (File S4 lists 17 records), which may well reveal different forms of Psittacanthinae pollen, or of less diverse New World lineages when re-studied using SEM.

Data-inherent shortcomings

The data assembled for our study from gene banks do not allow for conclusions at and below the genus level to be drawn. Genus-level data are limited, and in several cases where more than a single species (or individual) has been sequenced from the same genus, the genera do not show a high coherence when it comes to tree inferences (Fig. 7). This will become a problem when studying pollen grains from younger strata, which, increasingly, may show forms identical to one or more modern genera. For instance, our assessment of the Altmittweida MT is based on its similarity to the pollen of Amyema and Helixanthera figured in Grímsson, Grimm & Zetter (2017). In that study, material was used from vouchers identified as A. gibberula, the only species of the Amyeminae clade studied so far palynologically, and Helixanthera kirkii. According to our species-level analyses, species of neither of the two genera are resolved as sibling species. As exemplified in Fig. 7, the two or three sequenced species of Amyema are resolved at different placements in the Amyeminae subtree, but A. gibberula has not been sequenced at all. Helixanthera kirkii has only been sampled for nuclear data, and is placed far (phylogenetically speaking) from its congeners, which are scattered across the core Lorantheae subtree. Lacking any comparative data, it cannot be judged if these placements are genuine, or if one (or several) of the species (sequenced individuals) were misidentified/-associated (generic concepts are volatile in Loranthaceae, see synonymy lists provided by Tropicos.org, 2016). Thus, based on the available pollen of the Lorantheae and their established genetic affinities as members of the same clade, we can only assume with some certainty that the Altmittweida MT is a likely representative of the core Lorantheae, but not if it is a congener of Helixanthera, or more closely related to part of that genus. We also cannot judge to which degree Helixanthera pollen can be considered derived/unique enough within the core Lorantheae to warrant the association of a fossil pollen with a single extant genus.

Furthermore, we can only rely on fossil pollen of several northern hemispheric localities; localities we have been studying in the recent years. But most of the extant, and potentially extinct, diversity of Loranthaceae lies in the Southern Hemisphere (Figs. 10–11). South America, and in particular Africa, are much less studied palynologically than e.g. Europe, and the tradition of using SEM to study fossil pollen records is scant or absent in the Americas and Australasia (but see Ferguson et al., 2009; Bouchal, Zetter & Denk, 2016; del Carmen Zamaloa & Fernández, 2016). Nevertheless, there are records of Loranthaceae pollen from these areas, and if Antarctica is included (File S4), these records cover anything between the early Eocene and Holocene. Moreover, pollen assigned to Santalaceae or Viscaceae under LM may in fact be Loranthaceae Pollen Type A. Re-studying at least some of these assemblages using high-resolution SEM photography could provide much needed evidence for the distribution of different Loranthaceae lineages back in time. A more detailed and comprehensively studied pollen record at a global scale would also provide the necessary number of fossils to put forward and test explicit phylogeographic scenarios for the family. In the case of South America, particular fossil pollen can be straightforwardly compared to the substantial variation seen in the modern genera and species (seminal works of Feuer & Kuijt, 1979, 1980, 1985). It would be most interesting to pinpoint the earliest occurrences of the compact B-type pollen characteristic of the Cladocolea-Struthanthus lineage or the strongly derived C- and D-type pollen of the Passovia pyrifolia-Dendropemon-Orycthanthus clade. However, we are missing comprehensive molecular data on the Psittacanthinae at the intra-generic level and on species included in Passovia and Phthirusa (according Kuijt, 2011; see e.g. Fig. 7). A detailed molecular-phylogenetic framework would be necessary to depict evolutionary trends in pollen morphology of this group and to identify ancestral, more primitive (plesiomorphic) vs. modern, derived (apomorphic) pollen morphs of this lineage in the fossil record. Correlation of such data with palaeovegetational evidence (accompanying flora, in particular availability of mid- to high-canopy trees), may help to assess if the shift from root to aerial parasitism in currently exclusively aerial parasitic Loranthaceae lineages occurred before or after their establishment.

Due to the data-related limitations regarding both the molecular data and the fossil record, our dating analysis set-up can only provide absolute minimum estimates for divergence ages in the Loranthaceae. In a recent study on Osmundaceae, we observed that uncorrelated clock-inferred dates deviated from dates inferred with the recently proposed fossilised-birth-death dating approach (FBD; Heath, Huelsenbeck & Stadler, 2014), with the former tending to underestimate age (Grimm et al., 2015). In contrast to traditional node dating, FBD dating recruits the entire fossil record of a focal group and seems to outperform node dating in simulation and with real-world data (Heath, Huelsenbeck & Stadler, 2014; Grimm et al., 2015; Renner et al., 2016). In the case of Loranthaceae, the coverage of lineages with fossils and of the modern taxonomic diversity is insufficient for the application of FBD, although this approach would allow for a more appropriate handling of the fossils (including ours), namely as members of lineages, rather than minimum age priors for discrete MRCA. To avoid over-interpretation of the fossils during the latter, all fossil age constraints and estimates were used here in a conservative manner (see Descriptions; Inferences). More precise estimates and a larger taxon set would be needed to reconstruct explicit migration pathways of the different Loranthaceae lineages that consider the fossil record of the family.

Timing of evolution of main Loranthaceae lineages

The main, currently aerial parasitic lineages, of Loranthaceae evolved about 20 myrs earlier (Table 3) than estimated by Vidal-Russell & Nickrent (2008b); a discrepancy easily explained. In contrast to the earlier study, we can exclusively rely on ingroup fossils as age constraints, which provide direct evidence for the occurrence of several Loranthaceae lineages in the middle Eocene. Vidal-Russell & Nickrent (2008b) used two sets of fossil constraints for their dating of an all-Santalales dataset. The first set used a single fossil (Anacolosidites Cookson & K.Pike) to constrain the root age of an Olacaceae s.l. subclade, the former Anacolosideae (= Aptandraceae), to 70 Ma, providing generally older estimates than the second, preferred set. The second set used five additional fossils and included Cranwellia Sat.K.Srivast. to constrain the root age of Loranthaceae to >70 Ma. We again diverged from Vidal-Russell & Nickrent (2008b), by not using a different study, i.e. Wikström, Savolainen & Chase (2001), to constrain the (ingroup) root age. Using secondary dating constraints and age priors based on outgroup fossils typically leads to overly young age estimates (e.g. Grimm & Renner, 2013, for Betulaceae; Garzón-Orduña et al., 2015, for Solanaceae and Ithomiini; Schenk, 2016, for simulated data). For example, in the two families of Canellales, namely Canellaceae and Winteraceae, crown group estimates using ingroup fossils as age priors are about double the age of those inferred based on a large magnoliid dataset including only root age constraints for the Winteraceae and the order (Marquínez et al., 2009; Thomas et al., 2014; Massoni, Couvreur & Sauquet, 2015; Müller et al., 2015).

It must be noted that the existence of a lineage, as evidenced by the pollen record, does not allow for conclusions to be drawn regarding the parasitic habit of its extinct members. The Muellerina–Gaiadendron case shows that root and aerial parasites produce similar pollen grains. Even if we consider this pollen type to be primitive (‘symplesiomorphic’), the shift of the Lorantheae to aerial parasitism did not affect the pollen morphology in all of its sublineages to the same degree. The unconstrained topologies indicate several shifts from root to aerial parasitism within the family. It may thus be possible that more shifts occurred in the past than visible from the present-day situation. Ancient members of a Loranthaceae lineage may have been root parasites (or intermediate) in contrast to their modern representatives. Our older estimates nevertheless make sense considering the substantial genetic divergence between extant Loranthaceae, the backdrop of Cenozoic global climate evolution, and the evolutionary history of the potential hosts for aerial Loranthaceae: mid- to high-canopy trees (see also Fig. 8). Although some species of the Loranthaceae family seem to be linked to a specific host, the genera themselves usually parasitise a wide range of hosts, spanning different families and even orders (File S6). The colonisation potential of aerial mistletoes is high. For instance, the New Zealand endemic Ileostylus micranthus (Lorantheae: Ileostylinae) parasitises 47 different families, including northern hemispheric lineages introduced in historic times (Norton & de Lange, 1999). Australian mistletoes commonly infest two widespread, common and native tree genera (Acacia, Eucalyptus), but in total 256 genera are infested, and species of four genera can be found on exotic (introduced) tree genera such as Nerium, Quercus (oaks), Platanus, and Salix, among others (Downey, 1998). All these genera are potential hosts of northern hemispheric Loranthaceae (e.g. Loranthus europaeus), and can be traced back at least to the Eocene (e.g. Mai, 1995). For example, primary radiation and diversification of oaks—the most diverse, extratropical tree genus of the Northern Hemisphere with more than 400 modern species (Nixon, 1997; Huang, Zhang & Bartholomew, 1999)—was finished by the end of the Eocene (Hubert et al., 2014). The general vegetation types in which aerial Loranthaceae are found—various sorts of subtropical to temperate, non-frost forests but also tropical biomes—have been available through the entire Cenozoic (e.g. Mai, 1995). Most of the Eocene is characterised by a globally ameliorated climate (Zachos et al., 2001). During this time scale, tropical and subtropical forests reached a peak in their distribution, with subtropical and temperate forests reaching far north. This could have been the trigger for a global radiation of aerial parasites in Loranthaceae. In western Greenland, currently epiphytic Loranthaceae (Psittacanthinae; Aamaruutissaa MT, aff. Tripodanthus) co-occurred with a high variety of subtropical to temperate Fagaceae including various intrageneric groups of oaks (Grímsson et al., 2015). Fagaceae in general (see File S6) and oaks in particular are natural hosts of Eurasian Lorantheae. Oaks are major elements of extratropical northern hemispheric mid- to high-canopy forests and open woodlands. The Aamaruutissaa palynological assemblage covers representatives of ca. 30 families of woody angiosperms in total (Grímsson et al., 2014b), including many potential hosts of epiphytic Loranthaceae in modern-day extra-tropical North America and East Asia. The arborescent families Fagaceae, Juglandaceae, and Sapindaceae (including maples, Acer) can be found at all other localities included in our study (Table 1; File S7). All LM/SEM palynologically studied floras further comprise lianas (Vitaceae) and additional predominately or exclusively arborescent families such as Aquifoliaceae (Ilex), Cornaceae, Malvaceae, Myricaceae, Oleaceae, Platanaceae, and Ulmaceae. On the other hand, typically or exclusively herbaceous families are rare (or absent). Thus, the early Loranthaceae described here apparently thrived in densely forested habitats with ample niche opportunities for aerial parasites.

The mid-Oligocene falls into a phase of global cooling and retreat of subtropical and tropical vegetation belts to lower latitudes. If the main currently aerial parasitic lineages evolved during that time in Australia, as inferred by Vidal-Russell & Nickrent (2008a, 2008b; but see Barlow, 1990; Vidal-Russell & Nickrent, 2007), Loranthaceae would have needed to be extremely competitive to radiate at a global scale. With its (cold-)temperate to polar climate from the Oligocene onwards, Antarctica is an unlikely corridor for the global radiation of Loranthaceae. The situation in eastern North America and Europe, two areas heavily affected by the Pleistocene climate fluctuations, indicates that Loranthaceae cannot compete with their distant sister clade Viscaceae in the temperate zone, and there is no indication that any Loranthaceae lineage ever thrived in cold-temperate (boreal) climates. Long-distance dispersal via Africa or the Pacific is unlikely in the light of the modern distribution patterns (Fig. 11). All continental African species are members of the core Lorantheae, and distant relatives of the exclusively Australasian and South American lineages. The age estimates indicate that main Australasian (probably monophyletic Elytrantheae) and New World lineages (probably paraphyletic Psittacantheae) diverged around the same time (Fig. 8; Table 3), which would fit with the traditional Gondwana-Breakup scenario suggested for the family (Barlow, 1990; Vidal-Russell & Nickrent, 2007). Whether divergences in Loranthaceae are triggered by actual tectonic events has to be tested once a more comprehensive taxon and gene sample is available, and would require a re-investigation of the pollen record of the Southern Hemisphere using combined LM and SEM microscopy. With such data at hand, explicit pollen evolution scenarios could be established to discriminate between pollen indicative of ancestral or deep-rooting, slow-evolving (regarding their pollen morphologies) modern lineages, and extant genera or relatively late radiated supergeneric groups. The Oligocene cooling may have been the final trigger to isolate the American lineages from those in the Old World and Australasia. It also may have effected transcontinental exchange between Africa and East Asia, trigger the formation of the contemporary genera (Fig. 9; Table 3, but see the Discussion section), and manifest the isolation of Australasian lineages.

Conclusion

Molecular age estimates have often been criticised as being too young in comparison to the fossil record. The crown group radiation and associated onset of aerial parasitism in Loranthaceae, placed in the middle Oligocene by a study including all lineages of the Santalales (Vidal-Russell & Nickrent, 2008b), could have been taken for such a case. It would have invoked three difficult-to-understand phenomena: (1) Quick long-distance dispersal and rapid radiation on a global scale of a mostly tropical-subtropical lineage during a phase of global cooling; (2) Host-specialisation and simultaneous colonisation of subtropical forest elements that were already evolved by the Eocene, at least 20 myrs earlier; (3) The comparatively rich palynological record of the zoophilous Loranthaceae, with earliest reliable records in the Eocene of Australasia (south-eastern Australia, Tasmania), East Asia (Hainan, southern China), western Eurasia (Germany), the Americas (Argentina, southeastern United States) and Greenland reflect a largely lost diversity of root parasites or extinct sister lineages of extant Loranthaceae. These extinct lineages would then have been replaced, at the earliest, in the middle Oligocene (except for three refugia) in their entire range by their newly evolved aerial parasitic siblings. Using SEM-studied fossil pollen, we can push back the origin(s) of the main Loranthaceae lineages to at least the middle Eocene; a time when important hosts of modern epiphytic Loranthaceae evolved and radiated, and Earth enjoyed a phase of ameliorated climate. The new dating estimates are furthermore relatively stable regarding alternative rooting scenarios for the family.

Supplemental Information

Supplemental Information 1 Extended materials and methods.

This supplement details the steps of the data harvesting, curating, and analyses and includes an annotated tabulation of used node height priors for each scenario as well as comprehensive single-gene maximum likelihood trees inferred from the unfiltered harvested data.

Click here for additional data file.

Supplemental Information 2 Available and used sequence data of Loranthaceae.

Tabulation of used data; see sheet Content for further details.

Click here for additional data file.

Supplemental Information 3 Supplementary pollen plates.

This file includes Plates S01 to S16 referenced in the main text showing high-resolution LM and SEM micrographs (overviews and details) of the newly described fossil pollen.

Click here for additional data file.

Supplemental Information 4 Palaeopalynological record of Loranthaceae.

Palaeopalynological record compiled from literature and including the newly decribed finds. Unless noted otherwise in remarks column, each listed pollen has been checked and systematically assessed using the available documentation. See in-text Figs. 10 and 11 for maps of this list per time-slice.

Click here for additional data file.

Supplemental Information 5 Nonparametric bootstrap support for the comprehensive species sample.

Tabulation of (partly) competing split support using different gene samples, based on inferences done in course of analysis step 3 (see File S1). See also Splits-NEXUS files included in the according subfolder in the online supplementary archive (OSA).

Click here for additional data file.

Supplemental Information 6 Host specifity of loranths.

List of families infested by different loranth species compiled from various literature sources.

Click here for additional data file.

Supplemental Information 7 Accompanying palaeovegetation (family-level list) of here-in covered northern hemispheric sites comprising Loranthaceae fossils.

Click here for additional data file.

Supplemental Information 8 Online supplementary archive.

This archive includes all data and analysis files referred to in the main text and supplementary files that are needed to reproduce the shown results. See GuideToFiles.txt for naming schemes and contents of each subfolder included in this archive, and File S1 for analysis set-ups and further background information.

Click here for additional data file.

Ignacio Escapas and Benjamin Bomfleur are thanked for supplying South American literature, Kanchi Natarajan Gandhi from the IPNI team for clarifying the standard author form for the Indian palynologist Satish Srivastava, Marco Simeone for providing a compilation of plastid substitution rates reported in literature, and two anonymous reviewers and the editor for their constructive comments on the manuscript. Alastair Potts and Maxine Smit are acknowledged for proof-reading the original submission and final manuscript.

Additional Information and Declarations

Competing Interests

Author Contributions

The authors declare that they have no competing interests.

Friðgeir Grímsson conceived and designed the experiments, performed the experiments, contributed reagents/materials/analysis tools, wrote the paper, prepared figures and/or tables, reviewed drafts of the paper, and provided artwork (pollen plates); processed of samples and identification and taxonomic description of pollen types.

Paschalia Kapli conceived and designed the experiments, performed the experiments, analysed the data, contributed reagents/materials/analysis tools, reviewed drafts of the paper.

Christa-Charlotte Hofmann contributed reagents/materials/analysis tools, reviewed drafts of the paper, processed of samples and identification and taxonomic description of pollen types.

Reinhard Zetter contributed reagents/materials/analysis tools, reviewed drafts of the paper, processed of samples and identification and taxonomic description of pollen types.

Guido W. Grimm conceived and designed the experiments, performed the experiments, analysed the data, contributed reagents/materials/analysis tools, wrote the paper, prepared figures and/or tables, reviewed drafts of the paper, and created artwork (other than pollen plates); compiled online supporting archive.

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
