# Peer review of "Eocene Loranthaceae pollen pushes back divergence ages for major splits in the family"

_PeerJ, doi:10.7717/peerj.3373_

## Round 0.1 · original submission · Major Revisions

I really enjoyed your approach of integrating fossil data into disentangling the evolutionary history of plant parasitism. Working on parasitism myself, it is all too familiar how easily the fossil record is brushed away in favor of distribution data on extant parasites or their hosts (De Baets and Littlewood, 2015) – we could find little studies integrating fossil data for plants. I agree that you cannot see the extant diversity without data from the fossil record – certainly in groups with more readily fossilizable structures (e.g., pollen) where such data is available or easily obtainable. However, there are some points which need to be addressed before publication. I really see the importance and merit in this study and I therefore severely hope you will take the time to implement these suggestions. According to our guidelines and my assessment there is certainly no reason to reject this manuscript (as suggested by one of the reviewers). The main points to address are:

Phylogenetic analyses: It might be worthwhile to explore alternative methods such as maximum likelihood or maximum-parsimony methods and how missing data might affect your analysis (see comments by reviewer 1). You need to provide all details of the BEAST analysis and calibration strategies in the main text and it would be good to compare them with alternative methods of calibration or at least state why these were not used (see comments by reviewer 1).

Pitfalls of using general characteristics of pollen to infer aerial parasitism: pollen are highly resistant so more continuously available than complete plant body fossils making them interesting to study changes in composition over time. If pollen give no direct information on the type of parasitism (see comments by reviewers), Inference based on pollen morphology heavily relies on the taxonomic sampling of studies on morphology as well as available sequences of extant taxa which can occasionally be frustrating – even if you are adding a considerably amount of new data. I see no problem against using them - It is not uncommon in parasitic helminth to use eggs to be characteristic for certain species or even lineages, but the potential pitfalls should be highlighted (see also comments by reviewers 1 and 2). I suggest to explore alternative dating scenarios if assignment is ambiguous (e.g., line 310: see comments by reviewer 2) or at least highlight that this needs to be further tested and how (more sequence and/or data on pollen of extant taxa) in the discussion.

Break-Up of Gondwana: you discuss very little of how the evolution of would coincide with the break-up or biogeography in general, although you allude to it already in the abstract. Fragmentation of continents are often very complicated affairs and a lot could have happened after the Break-Up (De Baets et al., 2016). You only show a paleogeographic reconstruction which only gives an average view on the position of the coastline and climatic belts. These make certain assumptions to maximize land and/or sea (Smith, 2011; De Baets et al., 2016). I think you have two options. One option would be to reduce the speculation about the coincidence with the break-up of Gondwana to the discussion and designate that its needs to be tested further or leave it out entirely. I think it would be worthwhile to keep it in, but then you would need to discuss this in more detail and compare one split after another with the age constraints for the break-up of particular parts of Gondwana. A compromise might be too least ages of break-up in one of your graphs. I guess in theory, you could even use the pollen (of other plants) in your residues to decide if the first occurrence of certain Loranthaceae coincides with major changes in vegetation belts during, before or after the break-up to corroborate such hypotheses.

Justification of fossil age constraints: With your table, you fulfill the most important practices of fossil calibrations (Benton et al., 2009; Parham et al., 2012). However, you need to be more specific on the age evidence of your samples (see also comments by reviewer 1); what is the evidence for the age of each locality. I suspect mostly (bio)stratigraphic – so you would have to list the main evidence for a certain age (biozone) and correlate this with the latest geological time-scale for reproducibility (or at least list which one you used) for reproducibility. In the rare case, you have more precise absolute dates available you need to list these (but I suspect this not to be the case). This approach is explained in Benton et al. (2009); Parham et al. (2012); Sauquet et al. (2012). Some examples are also given in De Baets et al. (2016).

Personal communication: It is pointless to make a personal communication with an anonymous person; I am not a fan of personal communication with a non-anonymous person, but at least this can be tracked and verified. You can of course leave the information on how you performed the analysis without stating this relies on a personal communication with an anynomous person

Grimmson et al. 2017: You often refer to this publication which is in press but not yet published. We would have to wait to publish your manuscript until the manuscript is publicly accessible. I could not access the publication, but I suspect it should be soon available if it already received a doi number.

In addition to these and the points raised by the reviewers, please also take a look at my additional comments (see annotated manuscript). If you have any questions, please do hesitate to contact me.

Suggested references:

Benton, M. J., Donoghue, P. C. J., and Asher, R. J., 2009, Calibrating and constraining molecular clocks, in Hedges, B. S., and Kumar, S., eds., the Timetree of Life: Oxford, Oxford University Press, p. 35-86.
De Baets, K., Antonelli, A., and Donoghue, P. C. J., 2016, Tectonic blocks and molecular clocks: Philosophical Transactions of the Royal Society of London B: Biological Sciences, v. 371, no. 1699.
De Baets, K., and Littlewood, D. T. J., 2015, The Importance of Fossils in Understanding the Evolution of Parasites and Their Vectors: Advances in Parasitology, v. 90, p. 1-51.
Parham, J. F., Donoghue, P. C. J., Bell, C. J., Calway, T. D., Head, J. J., Holroyd, P. A., Inoue, J. G., Irmis, R. B., Joyce, W. G., Ksepka, D. T., Patané, J. S. L., Smith, N. D., Tarver, J. E., van Tuinen, M., Yang, Z., Angielczyk, K. D., Greenwood, J. M., Hipsley, C. A., Jacobs, L., Makovicky, P. J., Müller, J., Smith, K. T., Theodor, J. M., Warnock, R. C. M., and Benton, M. J., 2012, Best Practices for Justifying Fossil Calibrations: Systematic Biology, v. 61, no. 2, p. 346-359.
Sauquet, H., Ho, S. Y. W., Gandolfo, M. A., Jordan, G. J., Wilf, P., Cantrill, D. J., Bayly, M. J., Bromham, L., Brown, G. K., Carpenter, R. J., Lee, D. M., Murphy, D. J., Sniderman, J. M. K., and Udovicic, F., 2012, Testing the Impact of Calibration on Molecular Divergence Times Using a Fossil-Rich Group: The Case of Nothofagus (Fagales): Syst Biol, v. 61, no. 2, p. 289-313.
Smith, A. S., 2011, Uncertainties in Phanerozoic Global Continental Reconstructions and Their Biogeographical Implications, in Upchurch, P., McGowan, A. J., and Slater, C. S. C., eds., Palaeogeography and Palaeobiogeography: Biodiversity in Space and Time, CRC Press, p. 39-74.

Reviewer 1 ·

Basic reporting

The authors aimed to revisit the palaeopalynological record of Loranthaceae, using pollen ornamentation to discriminate lineages and to test molecular dating estimates about the origin of aerial parasitism in this Santalales family. For this aim they documented in detail, beautifully illustrated and analyzed fossil Loranthaceae pollen using sacanning-electron microscopy (SEM). These fossils, presumably from the Eocene and Oligocene, were associated with molecular-defined clades and used as minimum age constraints for Bayesian node dating using different topological scenarios. Their research endeavors to date the origin of aerial parasitism in Loranthaceae. Results of their analysis of fossil pollen data, phylogenetic analysis of molecular data downloaded from the GenBank using maximum-likelihood methods, and divergence-dating analysis using several calibration schemes identified the presence of at least one extant root-parasitic lineage (Nuytsieae) and two aerial parasitic lineages (Psittacanthinae and Loranthinae) by the end of the
Eocene in the Northern Hemisphere, using the fossil Loranthaceae pollen data for calibration of several nodes of the tree and using a molecular clock-based calibration of the tree root (Loranthaceae). Their results suggest that aerial parasitism in Loranthaceae evolved much earlier than previously suggested in the literature, and possibly it evolved multiple times. Taken together, their data point to the late Cretaceous-Paleogene continental breakup as the main trigger for initial diversification in Loranthaceae. The manuscript is in general well-written and the ideas are well-conceived. I believe this is an important addition to the rare comparative studies of pollen for this amazing plant family and that their results for using fossil pollen data in phylogenetic reconstruction would stimulate further study evaluating alternative historic scenarios in these mistletoes and particularly the origin of aerial stem-parasitism. However, there are some important points that must be corrected before this manuscript is recommended for publication in PeerJ. I offer broad areas of criticism and suggestions for improvement.
In addition, (1) I commend the authors for their extensive data sets (supplementary materials and SEM images and detailed descriptions of pollen data), compiled over many hours of lab work (SEM images) and computer work (data sets in supplementary materials). The SEM images and figures in general are high quality. In addition, the manuscript is clearly written in professional, unambiguous language though some areas of improvement are identified. If there is a weakness, it is in the phylogenetic and divergence-dating analyses (as I have noted below), which should be improved upon before acceptance of the manuscript. (2) I thank you for providing the curatorial data that accompany the DNA sequences downloaded from the GenBank, fossil records, supplementary plates of pollen grains, and host use data. Your supplemental files will be undoubtedly useful to future readers and certainly to dig-in for further research questions. Although your data sets and results are compelling, the data analysis should be improved in ways explained in other sections of this review.

Experimental design

Although your data sets and results are compelling, the data analysis should be improved in the following ways:

(1) Along with the maximum-likelihood method used in phylogenetic reconstruction, your analysis must be accompanied with the both Bayesian inference and maximum-parsimony methods. Although you provide an interesting analysis using each of the molecular markers to assess the effects of each in phylogenetic reconstruction, the data set combining all molecular markers identified as the best-phylogenetic estimate needs to be compared with those using Bayesian inference and maximum-parsimony methods. Please also provide an estimate of how missing data (lack of sequences in several markers) affect your phylogenetic reconstruction as opposed to a complete DNA sequence data matrix.

(2) Details of BEAST analysis and calibration strategies must be presented in the main text (incl. speciation model, clock model, tree priors, number of BEAST runs as replicates, burn-in %, effective samples sizes-ESS- values to assess the posterior distribution of all parameters and convergence among runs). Although you have conducted the divergence-dating analysis using several calibration strategies including a molecular clock-based calibration of the tree root, I am surprised that your dating analysis lacks comparisons with other strategies for tree calibration such as using mean substitution rates of each marker (i.e. geometric mean or instead of the use of absolute substitution rates, I would suggest sampling a range taking into account the standard deviation of the distribution of rates across species rather than a mean to take uncertainty into account), secondary calibrations from the literature + mean substitution rates, and/or combined with fossil records. I would be interested to know whether uncalibrated BEAST runs (i.e. without fossil pollen records) were consistent or inconsistent with the results reported here. Lastly, I am surprised that you apparently ignored other divergence date estimates and fossils available in the literature. In particular, the fossil record and previously established divergence times (secondary calibrations) between root parasites and aerial parasites could be used to calibrate the clock as alternative scenarios for comparison. The divergence time for the Loranthaceae clade from the Misodendraceae (81 ± 9 Ma; Vidal-Russell and Nickrent, 2008) could be used to calibrate the root node (normal distribution, mean 81, SD 9, range 90–72 Ma), and the age of 67.87 (normal distribution, mean 67.87, SD 6, range 75.38–65.62 Ma) to calibrate Loranthaceae (Magallón et al., 2015. New Phytologist 207:437–453). For the aerial parasites loranth crown group, the divergence between Nuytsia floribunda Australian root parasite and aerial loranth parasites clade could be used (Vidal-Russell and Nickrent, 2008b), approximating a median age of 28 Ma (normal distribution, mean 28, SD 6, range 34–22 Ma). The age of the fossil palynomorphs of Oryctanthus (Late Miocene to Mid-Pliocene; Graham, 1990) from Haiti could be used to set a minimum age of the extant Oryctanthus crown clade (3 Ma). For this calibration point you could use a lognormal prior, with the age of the Oryctanthus macrofossils (3 Ma) as the minimum age of the crown clade and the maximum of 6 Ma.

(3) The shortcomings of fossil pollen use in divergence-dating analysis must be acknowledged. Fossil pollen often has very high representation, especially for wind-pollinated species, and therefore, pollen produced by members of a given clade can appear in the fossil record relatively soon after the origin of that clade. However, fossil pollen is typically identified with relatively low taxonomic resolution, a problem exacerbated by the fact that the identification of fossil pollen types is rarely supported by synapomorphies and is often based solely on gross similarity. As a result, extinct plesiomorphic types can be confused with extant groups. This problem can be alleviated when fossil pollen data are accompanied with macrofossils or mesofossils (i.e., leaves, flowers and fruits). Please indicate whether or not these concerns apply to your study.

Validity of the findings

The next important issues regarding the validity of your findings include: (a) pollen descriptions, as far as I can grasp from my reading, lacks the needed associated stratigraphic information in terms of accurate correlation to the geologic timescale (GTS; Gradstein et al. 2004. 2004. A geologic time scale 2004. Cambridge, UK: Cambridge University Press). Please check the latest stratigraphic and geo-chronologic revisions of the rocks involved and mention explicitly the GTS of reference used in the calibration scheme (see further details on this issue in Sauquet et al. 2012. Syst. Biol. 61: 289–313). (b) the hypothesis testing, ages of divergence of Loranthaceae and origins of aerial-stem parasitism seem compromised by the apparently non-dated fossil pollen used for calibration of the Loranthaceae tree. I could not find the details for the absolute timescale (geo-chronology) used and the variation in the dates estimated for the substrates from which the pollen data were extracted from Please clarify.

Additional comments

The English language should be improved to ensure that your international audience clearly understands your text. I suggest that you have a native English-speaking colleague review your manuscript. Some examples where the language could be improved include lines 102–106, 107–108, 136–142, 655–661, 672–675, 744–747 – the current phrasing makes comprehension difficult.

Reviewer 2 ·

Basic reporting

The report of new pollen and their affinities to extant taxa is very important. However I have some issues in the analysis they used specially the rooting scenario they chose to discuss their results and shown in figure 9. It does not make sense with other supported published data and even they acknowledge that the fossil pollen from the Eocene has affinities with Nuytsia that is now consider as sister to all other Loranthaceae, I think the authors do not justify the use of Tupeia as sister to all Loranthaceae. Also they do not hypothesis any biogeographic they concentrate in the aerial parasitism origin in the family but they don’t have evidence on this just the pollen affinities to extant taxa there is a big assumption there.
I cannot access the article “Grímsson F, Grimm GW, Zetter R. 2017. Evolution of pollen morphology in Loranthaceae. Grana DOI:10.1080/00173134.2016.1261939.” I search in the journal site by that doi but I get the message: “Your search did not match any articles” and is not in the list of articles in issue 1 of 2017 either.
It is an important citation because the authors based their study much in that publication. For example the description of pollen types found in Loranthaceae.

Experimental design

The report of new pollen and their affinities to extant taxa is very important. However I have some issues in the analysis they used specially the rooting scenario they chose to discuss their results and shown in figure 9. It does not make sense with other supported published data and even they acknowledge that the fossil pollen from the Eocene has affinities with Nuytsia that is now consider as sister to all other Loranthaceae, I think the authors do not justify the use of Tupeia as sister to all Loranthaceae. Also they do not hypothesize any biogeographic scenario; they concentrate in the aerial parasitism origin in the family but they don’t have evidence on this just the pollen affinities to extant taxa. I think there is a big assumption there, how pollen type is link to aerial parasitism.

Validity of the findings

line 70
Thus, dispersed fossil pollen can aid in the reconstruction of past distribution of Loranthaceae lineages and shed light on the timing of the origin of the modern aerial parasitic clades.

The big assumption in this statement is that both characteristics are link. Why should they be?

Line 157
"The monotypic Tupeia is one of two Loranthaceae species with a spheroidal, echinate pollen as found in other Santalales lineages (Grímsson, Grimm & Zetter 2017) and the only one sequenced so far."
What about the other genus that has that pollen type as well but no sequence data? It is a big assumption to place Tupeia as sister to the other Loranthaceae based on pollen type and ignoring previous supported data and other evidences.

Line 163
which depicts the “correct relationships” between the major lineages
164 and potentially early diverging, isolated, monotypic genera (Anonymous, pers. comm., 2016)

Anonymous?

Line 310
Its morphology, place, and age would fit for an early precursor or extinct sister lineage of the Lorantheae

I don’t understand this statement because the author say this pollen morphotype is also similar to Nuytsia/Miller Clay Pit MT1, so why they assign it to an early precursor of Lorantheae?

Line 353
This sentences is not clear.
Notanthera heterophylla is of two species included in the two monotypic genera that comprise the South American Notantherinae, a subtribe of the Psittacantheae neither resolved as clade nor rejected with high support in molecular-phylogenetic inferences.

Line 588
In addition, we constrained our data to the topology of the Loranthaceae subtree as showed in Su et al. (2015; scenario 4), which—according to an expert on the group—is the most correct one to date (Anonymous, pers. comm., 2016; but see Grímsson, Grimm & Zetter 2017, file S6).

I think it is not necessary to cite the Anonymous comment, it does not add anything.

Line 808
…aerial parasitism in Loranthaceae evolved at least 20 myrs earlier (Table 3) than estimated by Vidal-Russell & Nickrent (2008b); a discrepancy easily explained.

I think this statement should be reformulated in Vidal-Russell & Nickrent 2008 the time estimation for aerial parasitism in Loranthaceae range from: 25 mya (± 5 mya) to 36 mya (± 11 mya) estimated by Bayesian relaxed molecular clock and by penalized likelihood from 32 to 38. These dates were extracted from table 2 of that publication.
What I can see that the estimation made by Vidal-Russell and Nickrent has more uncertainty that the one estimated by the authors of the present study, but the confidence intervals of both studies overlap.

Line 817
included Cranwellia Sat.K.Srivast. to constrain the root age of Loranthaceae to 70 Ma. [They write “for the crown group of Loranthaceae” in the text, p. 527, which, however, makes no sense regarding the corresponding results shown in table 2 on p. 531.]

I don’t understand why the authors say that results shown in table 2 of Vidal-Russell and Nickrent 2008 makes no sense with the calibration point chosen. Despite of the answer I think that statement does not add anything to the present study.

Fig 10 and 11 are the fossils reported in this study plus other studies? It in not clear.

Additional comments

The authors deal with aerial parasitism origins in the family; however, that is not the main aim of the manuscript and they don't give details how pollen grains are link to aerial parasitism. Fossil pollen grains related to extant mistletoe taxa could well have been root parasites in the past. This is a big assumption that the authors make with no justification. Aerial parasitism could have arise anywhere along the branch that leads to a clade in which extant taxa are branch parasites.

---

## Round 0.2 · Minor Revisions

Thank you for addressing all of our concerns. Your paper is as good as accepted. I still found some minor points i would like you to address before publication (see annotated manuscript).

---

## Round 0.3 · accepted · Accept

Thank you for making these final changes.